# The penultimate deglaciation: protocol for PMIP4 transient numerical simulations between 140 and 127 ka, version 1.0

Laurie Menviel[1*], Emilie Capron[2,3,*], Aline Govin[4], Andrea Dutton[5], Lev Tarasov[6], Ayako Abe-Ouchi[7], Russell N. Drysdale[8,9], Philip L. Gibbard[10], Lauren Gregoire[11], Feng He[12], Ruza F. Ivanovic[11], Masa Kageyama[4], Kenji Kawamura[13,14,15], Amaelle Landais[4], Bette L. Otto-Bliesner[16], Ikumi Oyabu[13], Polychronis C. Tzedakis[17], Eric Wolff[18], and Xu Zhang[19,20]

[1]Climate Change Research Center, PANGEA, the University of New South Wales, Sydney, Australia
[2]Physics of Ice, Climate and Earth, Niels Bohr Institute, University of Copenhagen, Tagensvej 8, DK-2100 Copenhagen Denmark
[3]British Antarctic Survey, High Cross, Madingley Road, Cambridge, CB3 0ET, UK
[4]Laboratoire des Sciences du Climat et de l'Environnement (LSCE), Institut Pierre Simon Laplace (IPSL), CEA-CNRS-UVSQ, Université Paris-Saclay, Gif-Sur-Yvette, 91190, France
[5]Department of Geological Sciences, University of Florida, PO Box 112120, Gainesville, FL 32611, USA
[6]Department of Physics and Physical Oceanography, Memorial University of Newfoundland, St John's, Canada
[7]Atmosphere and Ocean Research Institute, The University of Tokyo, Tokyo, Japan
[8]School of Geography, The University of Melbourne, Melbourne, Australia
[9]Laboratoire EDYTEM UMR CNRS 5204, Université Savoie Mont Blanc, 73376 Le Bourget du Lac, France
[10]Scott Polar Research Institute, University of Cambridge, Cambridge, CB2 1ER, UK
[11]School of Earth and Environment, University of Leeds, Leeds, LS2 9JT, UK
[12]Center for Climatic Research, Nelson Institute for Environmental Studies, University of Wisconsin-Madison, Madison, WI 53706, USA
[13]National Institute of Polar Research, Research Organizations of Information and Systems,10-3 Midori-cho, Tachikawa, Tokyo 190-8518, Japan
[14]Department of Polar Science, Graduate University for Advanced Studies (SOKENDAI), 10-3 Midori-cho, Tachikawa, Tokyo 190-8518, Japan
[15]Institute of Biogeosciences, Japan Agency for Marine-Earth Science and Technology, 2-15 Natsushima-cho, Yokosuka 237-0061, Japan
[16]Climate and Global Dynamics Laboratory, National Center for Atmospheric Research (NCAR), Boulder, CO 80305, USA
[17]Environmental Change Research Centre, Department of Geography, University College London, London, UK
[18]Department of Earth Sciences, University of Cambridge, Cambridge, CB2 3EQ, UK
[19]Alfred Wegener Institute, Helmholtz Centre for Polar and Marine Research, D-27570 Bremerhaven, Germany
[20]Key Laboratory of Western China's Environmental Systems (Ministry of Education), College of Earth and Environmental Sciences, Lanzhou University, Lanzhou, 730000, China
[*]Both authors contributed equally to this work

**Correspondence:** L. Menviel (l.menviel@unsw.edu.au) and E. Capron (capron@nbi.ku.dk)

**Abstract.** The penultimate deglaciation (PDG, ∼138-128 thousand years before present, hereafter ka) is the transition from the penultimate glacial maximum (PGM) to the Last Interglacial (LIG, ∼129-116 ka). The LIG stands out as one of the warmest interglacials of the last 800 ka, with high-latitude temperature warmer than today and global sea level likely higher by at least 6 meters. Considering the transient nature of the Earth system, the LIG climate and ice-sheets evolution were certainly influenced by the changes occurring during the penultimate deglaciation. It is thus important to investigate, with coupled

Atmosphere-Ocean General Circulation Models (AOGCMs), the climate and environmental response to the large changes in boundary conditions (i.e. orbital configuration, atmospheric greenhouse gas concentrations, ice-sheet geometry, and associated meltwater fluxes) occurring during the penultimate deglaciation.

A deglaciation working group has recently been set up as part of the Paleoclimate Modelling Intercomparison Project (PMIP) phase 4, with a protocol to perform transient simulations of the last deglaciation (19-11 ka; although the protocol covers 26-0 ka). Similar to the last deglaciation, the disintegration of continental ice-sheets during the penultimate deglaciation led to significant changes in the oceanic circulation during Heinrich Stadial 11 ($\sim$136-129 ka). However, the two deglaciations bear significant differences in magnitude and temporal evolution of climate and environmental changes.

Here, as part of the PAGES-PMIP working group on Quaternary Interglacials, we propose a protocol to perform transient simulations of the penultimate deglaciation under the auspices of PMIP4. This design includes time-varying changes in orbital forcing, greenhouse gas concentrations, continental ice-sheets as well as freshwater input from the disintegration of continental ice-sheets. This experiment is designed for AOGCMs to assess the coupled response of the climate system to all forcings. Additional sensitivity experiments are proposed to evaluate the response to each forcing. Finally, a selection of paleo records representing different parts of the climate system is presented, providing an appropriate benchmark for upcoming model-data comparisons across the penultimate deglaciation.

## 1  Introduction

Over the last 450 ka, Earth's climate has been dominated by glacial-interglacial cycles with a recurrence period of about 100 ka. These asymmetrical cycles are characterised by long glacial periods, associated with a gradual cooling, a slow decrease in atmospheric greenhouse gas (GHG) concentrations and a progressive growth of large continental ice-sheets in the Northern Hemisphere (NH), leading to a 60 to 120 m global sea-level decrease (Lisiecki and Raymo, 2005; Grant et al., 2014; Rohling et al., 2017). These long glacial periods were followed by relatively rapid multi-millennial-scale warmings into consecutive interglacial states. These deglaciations represent the largest natural global warming and large-scale climate reorganisations across the Quaternary. Deglaciations are paced by an external forcing, i.e. variations in the seasonal and latitudinal distribution of incoming solar radiation (insolation) driven by changes in Earth's orbit (Berger, 1978). However, changes in insolation alone are not sufficient to explain the amplitude of these major warmings and require amplification mechanisms. These amplification mechanisms are related to the large increase in atmospheric GHG concentrations (e.g. atmospheric $CO_2$ increases by 60 to 100 ppm (Lüthi et al., 2008)), the disintegration of NH ice-sheets and their associated change in albedo (Abe-Ouchi et al., 2013), as well as changes in sea-ice and vegetation cover (Fig. 1). Hence, deglaciations provide a great opportunity to study the interaction between the different components of the Earth System and climate's sensitivity to changes in radiative forcing.

A pervasive characteristic of the five deglaciations of the past 450 ka is the occurrence of millennial-scale climate events (e.g. Cheng et al., 2009; Barker et al., 2011; Vázquez Riveiros et al., 2013; Past Interglacials Working Group of PAGES, 2016; Rodrigues et al., 2017). In the North Atlantic, these events, also referred to as stadials, are characterised by a substantial

weakening of North Atlantic Deep Water (NADW) formation (e.g. McManus et al., 2004; Vázquez Riveiros et al., 2013; Böhm et al., 2015; Ng et al., 2018), possibly due to meltwater discharge into the North Atlantic (Ivanovic et al., 2018). There is a link between these events and enhanced iceberg calving, supported by the concurrent presence of Ice Rafted Debris (IRD) in North Atlantic marine sediment cores, and the stadials that contain substantial IRD layers (identified as Heinrich

Events) are known as Heinrich Stadials (e.g. Heinrich, 1988; Bond et al., 1992; McManus et al., 1999; van Kreveld et al., 2000; Hodell et al., 2017).

A weakening of the Atlantic meridional heat transport during these stadials maintains cold conditions at high northern latitudes in the Atlantic sector (Stouffer et al., 2007; Swingedouw et al., 2009; Kageyama et al., 2010, 2013) while contributing to a gradual warming at high southern latitudes (Blunier and Brook, 2001; Stocker and Johnsen, 2003; EPICA community members,

2006), thus leading to a bipolar-seesaw pattern of climate changes. The strengthening of NADW formation at the end of stadials induces a relatively abrupt temperature increase in the northern North Atlantic and surrounding regions, and a sharp increase in atmospheric $CH_4$ and Asian monsoon strength (Loulergue et al., 2008; Cheng et al., 2009; Buizert et al., 2014). While a significant atmospheric $CO_2$ increase is also observed during the NADW recovery at the end of Heinrich stadials during deglaciations (Marcott et al., 2014), the major phase of atmospheric $CO_2$ increase coincides with a Southern

Ocean (e.g. Barker et al., 2009; Uemura et al., 2018) and Antarctic warming (Cheng et al., 2009; Masson-Delmotte et al., 2011; Landais et al., 2013; Marcott et al., 2014) (Fig. 1). The sequence of events leading to the deglacial atmospheric $CO_2$ increase is still poorly constrained. Still, it most likely resulted from a combination of processes (e.g. Kohfeld and Ridgwell, 2009), including changes in solubility, global alkalinity content (e.g. Sigman et al., 2010), iron fertilization (e.g. Martin, 1990; Bopp et al., 2003; Martínez-García et al., 2014), Antarctic sea ice cover (Stephens and Keeling, 2000), and in ocean circulation

(e.g. Toggweiler et al., 2006; Anderson et al., 2009; Skinner et al., 2010; Toggweiler and Lea, 2010). Changes in ocean circulation, and particularly variations in the formation rates of the main deep and bottom water masses, i.e. NADW and Antarctic Bottom Water, can significantly impact atmospheric $CO_2$ by modifying the vertical gradient in oceanic dissolved inorganic carbon (e.g. Menviel et al., 2014, 2017, 2018).

While they share similarities, the last two deglaciations also bear significant differences in amplitude and durations. Figure 1

shows the evolution of key variables across 15 ka, from glacial maxima to peak interglacial conditions. The two deglaciations initiate under a range of glacial ice-sheet states and progress under a variety of orbital-forcing scenarios (Tzedakis et al., 2017; Past Interglacials Working Group of PAGES, 2016) (Fig. 1). Although there are still many open questions, the sequence of events occurring during the last deglaciation, which represents the transition from the Last Glacial Maximum (LGM, Marine Isotope Stage 2, hereafter MIS2, 26-19 ka) to our current interglacial, is starting to emerge (Cheng et al., 2009; Denton et al.,

2010; Shakun et al., 2012). The last deglaciation began with Heinrich Stadial 1 (HS1, ∼18-14.7 ka) when part of the Laurentide and Eurasian ice-sheets disintegrated (Dyke, 2004; Hughes et al., 2016), draining freshwater to the North Atlantic, that may have contributed to the observed weakening of NADW formation (e.g. McManus et al., 2004; Gherardi et al., 2009; Thornalley et al., 2011; Ng et al., 2018; Ivanovic et al., 2018). HS1 was characterised by cold and dry conditions in the North Atlantic, over Greenland and Europe (Tzedakis et al., 2013; Buizert et al., 2014; Martrat et al., 2014). Antarctic temperature

and global atmospheric $CO_2$ concentration rose during this period. Paleoproxy records and modelling studies suggest that the

Intertropical Convergence Zone (ITCZ) shifted southward (e.g. Arz et al., 1998; Chiang and Bitz, 2005; Timmermann et al., 2005; Stouffer et al., 2007; Kageyama et al., 2009, 2013; Marzin et al., 2013; McGee et al., 2014), thus leading to dry conditions in most of the Northern tropics, including the northern part of South America (Peterson et al., 2000; Deplazes et al., 2013; Montade et al., 2015) and the Sahel (Mulitza et al., 2008; Niedermeyer et al., 2009). Chinese speleothem records also indicate a weak East Asian summer monsoon activity during HS1 (Wang et al., 2001; Cheng et al., 2009).

The abrupt NADW resumption at $\sim$14.7 ka, imprinted by a sharp atmospheric $CH_4$ concentration increase (Loulergue et al., 2008), led to a warm period in the North Atlantic region, the Bølling-Allerød (Liu et al., 2009; Buizert et al., 2014) (Fig. 1, right). This period of North Atlantic warming coincides with (and may have triggered part of) the period of fastest sea-level rise, Meltwater Pulse-1A (MWP-1A) (Gregoire et al., 2016), during which sea-level rose by 14-18 m in less than 500 years, starting at $\sim$14.5 ka (e.g. Deschamps et al., 2012; Lambeck et al., 2014). At high southern latitudes, the gradual deglacial warming was interrupted by the Antarctic Cold Reversal (ACR, $\sim$14.5 - 12.8 ka) (Jouzel et al., 1995, 2007; Pedro et al., 2016), which was also coincident with a pause in the deglacial atmospheric $CO_2$ increase (e.g. Marcott et al., 2014). The ACR could be the result of enhanced northward heat transport from the Southern Hemisphere due to the strong NADW resumption occurring during the Bølling-Allerød (Pedro et al., 2016), or from a meltwater pulse originating from the Antarctic ice-sheet at the time of MWP-1A (Weaver et al., 2003; Menviel et al., 2011; Weber et al., 2014; Golledge et al., 2014).

A return to stadial conditions over Greenland, Europe and the North Atlantic occurred during the Younger Dryas ($\sim$12.8-11.7 ka, Fig. 1, right) (Alley, 2000). This event likely resulted from a combination of processes (Renssen et al., 2015), possibly including a weakening of NADW formation resulting from an increase in meltwater discharge into the Arctic Ocean (Tarasov and Peltier, 2005; Murton et al., 2010; Keigwin et al., 2018), or melting of the Fennoscandian ice-sheet (Muschitiello et al., 2015), and an altered atmospheric circulation due to a minimum in solar activity (Renssen et al., 2000). While the Barbados coral record suggests that a second phase of rapid sea-level rise occurred at about 11.3 ka (MWP-1B) (e.g. Bard et al., 1990), data from Tahiti boreholes (Bard et al., 2010) and from a compilation of sea-level data (Lambeck et al., 2014) provide no evidence of a particularly rapid sea-level rise during that period.

The penultimate deglaciation ($\sim$138-128 ka, referred here as PDG), which represents the transition between the PGM (MIS 6, also referred to as Late Saalian, 160-140 ka) and the LIG (also referred to as MIS 5e in marine sediment cores) (Govin et al., 2015), has received less attention. The PGM was characterised by an atmospheric $CO_2$ content of $\sim$195 ppm (Lüthi et al., 2008), and a significantly different extent of NH ice-sheets compared to the LGM (Dyke et al., 2002; Svendsen et al., 2004; Lambeck et al., 2006; Ehlers et al., 2011; Margari et al., 2014). The eustatic sea-level during the PGM is estimated at $\sim$90-100 m lower than present-day (Rabineau et al., 2006; Grant et al., 2012; Rohling et al., 2017), with a relatively large uncertainty range (Rohling et al., 2017). This compares to $\sim$130 m lower or more than during present-day during the LGM (Austermann et al., 2013; Lambeck et al., 2014). The LIG also bears significant differences to the interstadial that followed the last deglaciation, i.e. the Holocene. Latest data-based estimates suggest that sea-level was $\sim$6 to 9 m higher during the LIG than today, thus implying a significant ice-mass loss from both the Greenland and Antarctic ice-sheets (e.g. Dutton et al., 2015). In addition, compared to pre-industrial times, high-latitude SST and Greenland surface temperatures were respectively $\geq$1°C and 3 to 11°C greater during the LIG (e.g. Landais et al., 2016; Capron et al., 2017; Hoffman et al., 2017). Considering

the transient nature of the Earth system, a better understanding of the PDG could thus improve our knowledge of the processes that led to continental ice-mass loss during the LIG.

Recent work (Masson-Delmotte et al., 2011; Landais et al., 2013; Govin et al., 2015) also depicted a sequence of events over the PDG that contrasts with the one across the last deglaciation. Paleo-proxy records indicate that the disintegration of NH ice-sheets induced a ∼80 m sea-level rise (Grant et al., 2014) between 135 and 129 ka. This is also concomitant with Heinrich Stadial 11 (HS11) (e.g. Heinrich, 1988; Oppo et al., 2006; Skinner and Shackleton, 2006; Govin et al., 2015). HS11 was characterised by weak NADW formation (Oppo et al., 2006; Böhm et al., 2015; Deaney et al., 2017), cold and dry conditions in the North Atlantic region (Drysdale et al., 2009; Martrat et al., 2014; Marino et al., 2015), and gradually warmer conditions over Antarctica (Jouzel et al., 2007), associated with a sustained atmospheric $CO_2$ increase of ∼60 ppm between ∼134 and 129 ka (Landais et al., 2013) (Figs. 1, 2). Increasing evidence of sub-millennial scale climate changes at high and low latitudes during HS11 (e.g. Martrat et al., 2014) prompts the need to refine the sequence of events across the PDG. However, this is challenging as (i) climatic reconstructions over the PDG are still scarce and most records have insufficient resolution to allow identification of centennial- to millennial-scale climatic variability; and (ii) it is difficult to establish robust absolute and relative chronologies for most paleoclimatic records across this time interval (Govin et al., 2015).

While our knowledge of the processes and feedbacks occurring during deglaciations has significantly improved over the last two decades (e.g. Cheng et al., 2009; Shakun et al., 2012; Abe-Ouchi et al., 2013; Landais et al., 2013; Cheng et al., 2016), many unknowns remain. For example, our understanding of the precise roles of atmospheric and oceanic processes in leading to the waning of glacial continental ice-sheets during deglaciations is still incomplete. It is also crucial to comprehend the subsequent impacts of continental ice-sheets disintegration on the oceanic circulation, climate, the terrestrial vegetation and the carbon-cycle system.

Numerical simulations performed with climate models provide a dynamical framework to understand the response of the Earth system to external forcing (i.e. insolation) and internal dynamics (e.g., albedo, GHGs) that culminate in deglaciations. Atmospheric and oceanic teleconnections associated with millennial-scale variability can also be studied in detail. Model-paleoclimate proxy comparisons, including snapshot experiments at 130 ka and at 126 ka, suggest that the inclusion of freshwater forcing in the North Atlantic due to the melting of NH ice-sheets could explain the relatively cold conditions in the North Atlantic and warm conditions in the Southern Ocean observed in paleodata during these time periods (Govin et al., 2012; Stone et al., 2016). However, snapshot experiments assume that the climate state is in near-equilibrium, and because relatively rapid and large changes in both internal and external forcings occur during deglaciations, transient simulations (i.e. numerical simulations with time-varying boundary conditions) are needed. These simulations also allow a more robust paleodata-modelling comparison, thus enabling the refinement of the sequence of events.

Transient numerical simulations have already been performed for the last deglaciation (Liu et al., 2009; Menviel et al., 2011; Roche et al., 2011; Gregoire et al., 2012; He et al., 2013; Otto-Bliesner et al., 2014) and provide a dynamical framework to further our understanding of the climate-change drivers, teleconnections and feedbacks inherent in the Earth system. Currently, no 3-dimensional transient simulations were run across the full time interval covered by the penultimate deglaciation (∼140-127ka). However, transient simulations covering the period 135 to 115 ka have been performed with a range of models to

understand the impact of surface boundary conditions and freshwater fluxes on the LIG (Bakker et al., 2013; Loutre et al., 2014; Goelzer et al., 2016a). The need for transient simulations is now recognised and a protocol to perform transient experiments of the last deglaciation as part of PMIP4 has been recently established (Ivanovic et al., 2016). However, to further our understanding of the processes at play during deglaciations, including the role of millennial-scale climate change, other deglaciations should be studied in detail. Transient simulations of the penultimate deglaciation could also help to understand the climate and sea-level high-stand occurring during the LIG. We thus propose to extend the PMIP4 working group on the last deglaciation to include the penultimate deglaciation and thus create a Deglaciation working group. This effort will complement the last deglaciation experiments of PMIP4 (Ivanovic et al., 2016), allowing an evaluation of the similarities and differences in the climate system response during the last and the penultimate deglaciations.

Here we present a protocol to perform transient numerical simulations of the PDG from 140 to 127 ka with coupled AOGCMs. These experiments will provide a link with the PMIP4 transient LIG experiment (127 to 121 ka) (Otto-Bliesner et al., 2017). After a description of changes in insolation (section 2), GHGs (section 3), continental ice-sheets (section 4), and sea-level (section 5) occurring during the PDG, we present a framework to perform transient simulations of PDG (sections 6 and 7), as well as a selection of key paleoclimate and paleoenvironmental records to be used for model/data comparisons (section 8).

## 2 Insolation

The orbital parameters (eccentricity, obliquity, and longitude of perihelion) should be time evolving and set following Berger (1978). This external forcing affects the seasonal and latitudinal distribution, as well as the magnitude of solar energy received at the top of the atmosphere and, in the case of obliquity (Earth's axial tilt), the annual mean insolation at any given latitude with opposite, but small, effects at low and high latitudes. Eccentricity is high during the entire deglaciation period, ranging from 0.033 at 140 ka to 0.041 at 127 ka, and is significantly higher than during the last deglaciation (Fig. 1b; ∼0.019 at the LGM to 0.020 at 14 ka), and than the present value of 0.0167. Obliquity peaks at 131 ka; the degree of tilt is similar between the last and the penultimate deglaciation. Perihelion occurs near the NH winter solstice at 140 ka, shifting to near the NH summer solstice by 127 ka.

Although the overall trends in summer solstice insolation anomalies, as compared to the mean of the last 1000 years, evolve similarly in the last and penultimate deglaciations, the magnitudes of the maximum positive summer anomalies in the NH and the minimum negative summer anomalies in the Southern Hemisphere are much greater during the PDG, when eccentricity is higher, than during the last deglaciation (Fig. 1a). At 65°N, peak summer anomalies of more than 70 W/m$^2$ occur at 128 ka. In contrast, during the last deglaciation, the 65°N summer solstice anomalies peak at 11 ka, with anomalies of ∼50 W/m$^2$. Similarly, 65°S summer solstice negative anomalies are close to -40 W/m$^2$ at 127 ka, but only about -20 W/m$^2$ at 10 ka. In addition, the rates of change of summer solstice anomalies are greater from 140 to 127 ka than from 21 to 8 ka.

Given the clear differences between the solar forcing of last and penultimate deglaciations, comparing the two transient deglacial simulations will provide valuable information on the underlying mechanisms and Earth system feedbacks. The solar

constant should be set to 1360.7 W/m$^2$, consistent with the CMIP6-PMIP4 *piControl* and *lig127k* simulations (Otto-Bliesner et al., 2017) as well as the PMIP4 transient climate simulation of the last deglaciation (Ivanovic et al., 2016).

## 3 Greenhouse gases

GHG records are available solely from Antarctic ice cores across the time interval 140-127 ka (Fig. 2). LIG GHG records from the NEEM and other Greenland ice cores are affected by stratigraphic disturbances and in-situ $CO_2$, $CH_4$ and $N_2O$ production (e.g. Tschumi and Stauffer, 2000; NEEM community members, 2013). The NGRIP ice core provides a continuous and reliable $CH_4$ record but it only extends back to ∼123 ka (North Greenland Ice Core project members, 2004). We first briefly describe existing atmospheric $CO_2$, $CH_4$ and $N_2O$ records (below). The recent spline-smoothed GHG curves calculated from a selection of those records (Köhler et al., 2017) should be used. They have the benefit to provide continuous GHG records, with a temporal resolution of 1 yr on the commonly-used AICC2012 gas age scale (Bazin et al., 2013; Veres et al., 2013). This time scale is associated with an average 1$\sigma$ absolute error of ∼2 ka between 140 and 127 ka.

Atmospheric $CO_2$ concentrations have been measured on the EDC and TALDICE ice cores (Fig. 2). The EDC records from Lourantou et al. (2010) and Schneider et al. (2013) agree well overall. The Schneider et al. (2013) dataset depicts a long-term $CO_2$ increase starting at ∼137.8 ka and ending at ∼128.5 ka with a centennial-scale $CO_2$ rise above the subsequent LIG $CO_2$ values, also referred to as an "overshoot". The $CO_2$ overshoot is smaller in the Schneider et al. (2013) dataset compared to a similar feature measured in Lourantou et al. (2010): while the former displays a relatively constant $CO_2$ concentration of ∼275 ppm between 128 and 126 ka, the latter shows a $CO_2$ decrease from 280 to 265 ppm between 128 and 126 ka. The offsets between $CO_2$ records from the same EDC core are likely related to the different air extraction techniques used in the two studies (Schneider et al., 2013). The smoothed spline $CO_2$ curve, that should be used as forcing for the PDG is based on those two EDC dataset and the calculation method accounts for such potential difference in local maxima (details provided in Köhler et al. (2017)).

Atmospheric $CH_4$ concentration records from Vostok, EDML, EDC and TALDICE agree well within the gas-age uncertainties attached to each core (Fig. 2). They illustrate a slow rise from ∼390 to 540 ppb between ∼137 ka and 129 ka that is followed by an abrupt increase of ∼200 ppb reaching maximum LIG values at ∼128.5 ka. Because $CH_4$ sources are located mostly in the NH, an interpolar concentration difference (IPD) between Greenland and Antarctic $CH_4$ records exists.

For instance, an IPD of ∼14 ppb, ∼34 ppb and ∼43 ppb is reported during the LGM, Heinrich Stadial 1 and the Bølling warming respectively (Baumgartner et al., 2012). However, without reliable $CH_4$ records from Greenland ice cores, it remains challenging to estimate the evolution of the IPD across the deglaciation. Hence, for the atmospheric $CH_4$ forcing of transient PDG simulations, the smoothed spline $CH_4$ curve, which is solely based on the EDC $CH_4$ record (Köhler et al., 2017) should be used, recognising that the values may be 1-4% lower than the actual global average.

Both $CO_2$ and $CH_4$ concentrations undergo some rapid changes around 140 ka, which is also the time when the models should spin-up. To avoid possible artificial abrupt changes in the GHG, the average values obtained for the interval 139-141 (i.e. 191 ppm for $CO_2$ and 385 ppb for $CH_4$, Table 1) should be used as spin-up $CO_2$ and $CH_4$ concentrations. Consequently,

CO$_2$ and CH$_4$ changes between 140 and 139 ka provided in the forcing scenarios are linearly interpolated between the 140 ka spin-up values and those at 139 ka of 196.68 ppm for CO$_2$ and 287.65 ppb for CH$_4$. From 139 ka, the spline-smoothed curves from Köhler et al. (2017) should be used.

Atmospheric TALDICE, EDML and EDC N$_2$O records are available between 134.5 and 127 ka (Fig. 2) (Schilt et al., 2010;
Flückiger et al., 2002). From 134.5 to 128 ka, N$_2$O levels increase from ~220 to 270 ppb. Following a short decrease until ~127 ka, N$_2$O concentrations stabilise afterwards. No reliable atmospheric N$_2$O concentrations are available beyond 134 ka as N$_2$O concentrations measured in the air trapped in ice from the PGM are affected by in-situ production related to microbial activity (Schilt et al., 2010). During the LGM (considered here as the time interval 26-21 ka), the average N$_2$O level was ~201 ppb. Assuming the LGM is an analogue for the PGM, we propose a 140 ka spin-up value and N$_2$O transient forcing curve that
starts with a 201 ppb level and then linearly increases to 218.74 ppb at 134.5 ka. From 134.5 ka, the N$_2$O smoothed spline curve calculated by Köhler et al. (2017), which is based on the TALDICE and EDC discrete N$_2$O measurements, should be used.

The CO$_2$ and N$_2$O levels from the spline curves at 127 ka (274 ppm and 257 ppb) only differ from the values chosen as boundary conditions for the PMIP4 *lig127k* equilibrium experiment by 1 ppm and 2 ppb respectively (Otto-Bliesner et al.,
2017; Köhler et al., 2017). The comparison is less direct for CH$_4$. Indeed a global CH$_4$ value (685 ppm) rather than an Antarctic ice core-based CH$_4$ value (e.g. CH$_4$ level of 660 ppm at 127 ka in Köhler et al. (2017)) is proposed as forcing for the *lig127k* simulations. However, this difference in global atmospheric CH$_4$ and Antarctic ice core CH$_4$ concentration is similar to the one observed during the mid-Holocene (23 ppb) (Otto-Bliesner et al., 2017; Köhler et al., 2017).

## 4  Continental ice-sheets

Changes in continental ice-sheets during the deglaciation will significantly impact the climate system through their albedo, which will directly affect the radiative balance (e.g. He et al., 2013). Changes in continental ice-sheets geometry can also significantly impact atmospheric dynamics (e.g. Zhang et al., 2014; Gong et al., 2015). Transient simulations of the PDG will thus need to be forced by the 3-dimensional and time-varying evolution of continental ice-sheets, that is currently only available from numerical simulations. However, simulating the evolution of continental ice-sheets across the PDG is associated with large
uncertainties, due to the climate forcing of the ice-sheet models and poorly constrained non-linearities within the ice-sheet system. Glacial geological data (e.g. glacial deposits, glacial striations...) are also available to constrain continental ice-sheet evolutions and can thus provide an estimate of the uncertainties associated with the numerical ice-sheet evolutions. In this section, we describe the available numerical ice-sheet evolutions to use as a forcing of the transient simulations of the PDG. We further compare the results of these simulations with existing glacial geological constraints.

### 4.1  Combined ice-sheet forcing

To facilitate the transient simulations of the PDG, we are providing a combined ice-sheet forcing (available in the Supplement and on the PMIP4 wiki), in which separate reconstructions of different ice-sheets have been merged. As the sea-level solver

assumes an equilibrium initial condition, the simulations start at the previous interglacial. As is standard, the solver also requires present-day ice-sheet histories to bias correct against present-day observed topography. Thus, a full 240 ka ice-sheet history is required. The simulated NH ice-sheet evolution, described in Section 4.2 (Abe-Ouchi et al., 2013), is merged with the simulated Greenland and the Antarctic (Briggs et al., 2014) evolutions described in Sections 4.3 and 4.4, respectively. The resolution of the merged ice-sheet file is 1° longitude by 0.5° latitude. The merger involves no extra smoothing, beyond that inherent in the glacial isostatic adjustment solver, which involves transformation to spherical harmonics. The merger involves a simple masking operation with the mask boundary through Nares Strait, Baffin Bay, Davis Strait, and the Labrador Sea. Examination of the resultant topography shows small merger artifacts around Nares Strait ranging to a few hundred metres in elevation difference.

From the LIG onward, the combined ice-sheet evolution, referred to as GLAC-1D in PMIP4, is used (e.g. Ivanovic et al., 2016). GLAC-1D includes the Greenland and Antarctic ice-sheets components described in section 4.3 and 4.4 (Briggs et al., 2014), the North American ice-sheet simulation described in Tarasov et al. (2012) and the Eurasian ice-sheet simulation described in Tarasov (2014). The ice-sheet thickness from these simulations are run through a sea-level solver using the VM5a (Peltier and Drummond, 2008) Earth rheology to extract a gravitationally self-consistent topography. The surface topography is then run through a global surface drainage solver (using the algorithm described in Tarasov and Peltier, 2006) to extract the relevant surface drainage pointer field for each time-slice. This will indicate in which ocean grid cell each terrestrial grid cell will drain into.

## 4.2 North American and Eurasian ice-sheets

The evolution of NH ice-sheets during the PDG is given by a numerical simulation performed with the thermo-mechanically coupled ice-sheet model IcIES (Ice-sheet model for Integrated Earth system Studies) with an original resolution of 1° by 1° in horizontal and 26 vertical levels (Abe-Ouchi et al., 2007) (Fig. 3). IcIES uses the shallow ice approximation and computes the evolution of grounded ice but not floating ice shelves. The sliding velocity is related to the gravitational driving stress according to Payne (1999) and basal sliding only occurs when the basal ice is at the pressure melting point. This ice-sheet model was driven by climatic changes obtained from the MIROC GCM (Abe-Ouchi et al., 2013), which was forced by changes in insolation and atmospheric $CO_2$ concentration. In global agreement with glacial geological constraints (Dyke et al., 2002; Svendsen et al., 2004; Curry et al., 2011; Syverson and Colgan, 2011) and other numerical simulations of NH ice-sheets evolution (Tarasov et al., 2012; Abe-Ouchi et al., 2013; Peltier et al., 2015; Colleoni et al., 2016), the simulated extent and volume of the North American ice-sheet was smaller during the PGM than the LGM (Fig. 3).

In Eurasia, the PGM recorded the most extensive glaciation since MIS 12 (Hughes and Gibbard, 2018). The maximum extent of the Fennoscandian ice-sheet probably occurred at ∼160 ka, when it extended into central Netherlands, Germany, and the Russian Plain (Margari et al., 2010; Ehlers et al., 2011; Hughes and Gibbard, 2018). This was followed by a partial melting of the Fennoscandian ice-sheet, peaking between ∼157-154 ka, and a readvance after 150 ka (Margari et al., 2010; Hughes and Gibbard, 2018). The maximum extent of the NH ice-sheets probably occurred at the end of the PGM (Margari et al., 2014; Head and Gibbard, 2015), In Europe the late PGM ice advance was less expensive than at ∼160 ka, but

that was compensated by ice-sheet expansion in Russia, Siberia (Astakhov et al., 2016) and in North America (e.g. Curry et al., 2011; Syverson and Colgan, 2011). Glacial geological constraints (e.g. Astakhov, 2004; Svendsen et al., 2004) indeed suggest that the Barents-Kara ice-sheet extended further during the PGM than the LGM. The simulated Eurasian ice-sheet is in general agreement with the reconstruction of Lambeck et al. (2006), with a dome reaching 3000 m over the Kara Sea during the PGM that subsequently disintegrated across the deglaciation. However, the extent and volume of the simulated Eurasian ice-sheet might be underestimated since it is smaller at the PGM than LGM, whereas reconstructions suggest it should be larger at the PGM (Lambeck et al., 2006; Rohling et al., 2017).

Rohling et al. (2017) further suggest that the ice volume was almost equally distributed between Eurasia and North America at the PGM, with a 33 to 53 m global mean sea level equivalent (sle) contribution from the Eurasian ice-sheet and 39-59 m from North America, whereas the ice-sheet simulation produces a ∼24 m sle contribution from Eurasia. Thus, the volume of the North American ice-sheet may also be overestimated.

In the ice-sheet simulation, NH ice-mass loss follows closely the boreal summer insolation and occurs mostly between ∼134 and 127 ka (Fig. 4), with two peaks of glacial meltwater release at ∼131 ka and 128 ka. By 132 ka, the Eurasian ice-sheet has decreased significantly and the southern and western flanks of the North American ice-sheet have disintegrated (Fig. 3). Another significant retreat of the North American ice-sheet occurs between 132 and 128 ka, at which point it is mostly restricted to the north of the Hudson Bay. By 127 ka, the North American ice-sheet only remains over Baffin Island.

### 4.3 Greenland ice-sheet

The Greenland model uses an updated version (GSM.G7.31.18) of the Glacial Systems Model (e.g. Tarasov et al., 2012) run at grid resolution of 0.5° longitude by 0.25° latitude. The model has been upgraded to hybrid shallow-ice and shallow-shelf physics, with ice dynamical core from Pollard and DeConto (2012) and includes: a 4 km deep permafrost resolving bed thermal component (Tarasov and Peltier, 2007), visco-elastic bedrock response with global ice-sheet and sea-level loading, sub-shelf melt, parametrizations for subgrid mass-balance and ice flow (Morzadec et al., 2015), and updated parametrizations for surface mass-balance and ice calving.

Given that with active bed thermodynamics (down to 4km), the thermodynamic equilibration timescale is greater than 100 ka for the Greenland ice-sheet, the most appropriate method is to start the run during the previous interglacial period. Therefore, the model runs start at 240 ka with present-day ice and bedrock geometry and with an ice and bed temperature field from the end of a previous 240 ka model run. The model is then forced from 240 ka until 0 ka, with a climate forcing that is partly glacial index based, using a composite of a glaciological inversion of the GISP II regional temperature change (for the last 40 ka) and the synthetic Greenland $\delta^{18}O$ curve that was deduced from the Antarctic EDC isotopic record assuming a thermal bipolar seesaw pattern (Barker et al., 2011). The climate forcing also includes a 2-way coupled 2D energy balance climate model (Tarasov and Peltier, 1997) to capture radiative changes.

Greenland ice-sheet model runs are scored against a large set of constraints including relative sea level (RSL), proximity to present-day ice-surface topography, present-day observed basal temperatures from various ice cores, time of deglaciation of Nares Strait, and the location of the present-day summit. The last 20 ka of the run is critical as this represents the time

period with most of the data constraints for Greenland. The simulation presented here (G9175) is a least misfit model from a preliminary exploratory ensemble.

This simulation suggests no significant change in Greenland ice-mass until ∼134 ka (Fig. 4), followed by a small ice-mass loss, mostly from floating ice, between 134 and 130 ka. In this simulation, the main phase of Greenland deglaciation occurs
between 130 and 127 ka, during which Greenland loses an ice mass of 2.9 m sle in excess of the total pre-industrial value, and then an additional 1.5 m sle. As shown in Figure 5, the extent and height of the Greenland ice-sheet is significantly smaller at 128 ka than at 132 ka. Greenland ice-mass loss is particularly evident on its western side, with a part of southwestern Greenland being ice-free. To a first order, the simulated disintegration of the Greenland ice-sheet follows the increase in boreal summer insolation and in atmospheric $CO_2$ (Fig. 1).
The main phase of the Greenland ice-sheet retreat in this simulation is globally in agreement with proxy records, which suggest significant runoff in the Labrador Sea at ∼130 ka and at ∼127 ka (e.g. Carlson and Winsor, 2012). However the simulated Greenland ice-sheet disintegration could be too rapid as paleoproxy records suggest significant meltwater discharge from the Greenland ice-sheet throughout the LIG (e.g. Carlson and Winsor, 2012). In addition other model simulations suggest a maximum sea-level contribution from Greenland at ∼123-121 ka (Yau et al., 2016; Bradley et al., 2018), in agreement with
the timing of the LIG minimum elevation at the Greenland NEEM location. This minimum elevation estimate was reconstructed from total air and water isotopic records measured on the deep ice core drilled at that site (NEEM community members, 2013). Based on the paleoproxy records and model simulations, the protocol for the PMIP4 LIG simulation for 127ka (*lig127k*) recommends a pre-industrial Greenland configuration.

### 4.4 Antarctic ice-sheet

The Antarctic model configuration is largely that of Briggs et al. (2013, 2014): a hybrid of the Penn State University ice-sheet model (Pollard and DeConto, 2012) and the GSM. Simulations are run with a 40-km grid resolution using the LR04 benthic $\delta^{18}O$ stack (Lisiecki and Raymo, 2005) for sea-level forcing. The climate forcing is a parametric mix of an index based approach (using the EDC $\delta D$ record of Jouzel et al. (2007)) and one based on orbital forcing, as detailed by Briggs et al. (2013). The parameter vector (nn4041 from Briggs et al., 2014) that gave the best fit to constraints (GSM-A) in the large
ensemble analysis is used. Two changes are imposed on the model to partially rectify an inadequate LIG sea-level contribution. First, SST dependence is added to the sub-shelf melt model. Second, to compensate for inadequate LIG warming, where SSTs are above present-day values, they are then given a minimum value of 3°C (i.e. SST=MAX(SST,3.0°C)). Even so, the Antarctic contribution to the LIG high-stand is only 1.4 m sle and is therefore inadequate given current inferences (as well as constraints on contributions from Greenland and steric effects) (e.g. Kopp et al., 2009).
The simulation suggests a continuous Antarctic ice-sheet discharge during the PDG, with a glacial ice-mass loss of ∼12.5 m sle between 140 and 131 ka, followed by an additional 1.4 m sle between 131 and 130 ka (Fig. 4). In this simulation, the West Antarctic ice-sheet loses significant ice-mass between 140 and 136 ka (Fig. 6), with a retreat of the grounding line over the Ross Sea as well as ice-mass loss in the Weddell Sea, on the Antarctic Peninsula and in the Amundsen Sea sector. By 132 ka, the grounding line has completely retreated over the Ross Sea and has retreated significantly over the Weddell Sea.

## 5 Sea-level

Direct evidence for constraining the evolution of the global sea level during the time interval 140-127 ka remains sparse. Although the LR04 benthic $\delta^{18}O$ stack (Lisiecki and Raymo, 2005) is sometimes used to approximate sea-level change on glacial-interglacial timescales, in the case of the PDG, the timing of the LR04 benthic $\delta^{18}O$ stack is fixed by reference to a

handful of U-series coral dates from Huon Peninsula with relatively high analytical uncertainties and questionable preservation (Bard et al., 1990; Stein et al., 1993). Tying the MIS 5e peak to the average age of these coral dates results in a benthic $\delta^{18}O$ minimum that is roughly centred on the main phase of coral growth during this interglacial period (122 - 129 ka) (Stirling et al., 1998) rather than having the onset of the interglacial aligned with the timing of the onset of the sea level high-stand at far-field sites ($\sim$129 ka) (e.g. Stirling et al., 1998; Dutton et al., 2015).

Here, we seek to provide an improved reconstruction of sea level across the PDG by examining available RSL records. Information on the timing and magnitude of the changes across this time interval is provided by three RSL records (Fig. 7):

*i)* A RSL record from the Red Sea (Grant et al., 2012), that is deduced from the planktic foraminifera $\delta^{18}O$ measured on sediment cores retrieved in this evaporative marginal sea. This record is transformed into a RSL signal by using hydraulic models that constrain the salinity of surface waters as a function of sea level. The Red Sea record provides the only continuous

profile of RSL across our interval of interest;

*ii)* RSL data from the U-series dates and elevations of the submerged coral reefs of Tahiti (Thomas et al., 2009), and

*iii)* RSL data derived from U-series dates and elevations of uplifted coral terraces of Huon Peninsula, Papua New Guinea (Esat et al., 1999).

Providing a robust age model for sediment records from the PGM to the LIG is not straightforward (e.g. Govin et al., 2015)

and over time, several age models have been proposed for the Red Sea RSL record (e.g. Siddall et al., 2003; Rohling et al., 2009; Grant et al., 2012). The latest chronology is based on climatic alignment of the Red Sea RSL record to eastern Mediterranean planktic foraminifera $\delta^{18}O$ records, which are in turn aligned onto the absolutely-dated Soreq cave speleothem $\delta^{18}O$ record (Grant et al., 2012). While the absolute ages of the speleothem record have the potential to provide a more robust age model (both in terms of accuracy and precision), the application of these dates to the Red Sea sea level reconstruction hinges

on the assumption that the tie points between the Red Sea and eastern Mediterranean records have been correctly assigned, and that the intervals between these tie points can be linearly extrapolated.

The Tahiti and Huon Peninsula corals are associated with absolute radiometric dates (using U-series geochronology). For the purpose of this study, all of the U-series ages have been recalculated to normalize them with the same set of decay constants for $^{234}U$ and $^{230}Th$ (Cheng et al., 2013) (Tables S3 and S4), using the methodology described by Hibbert et al. (2016). The

array of data from Huon Peninsula suggest post-depositional alteration (open-system behaviour of the U-series isotopes) that complicates a precise age interpretation (Fig. 7).

The Red Sea time series published by Grant et al. (2012) depicts that, after a RSL low stand of about -100 m relative to present between 145 and 141 ka, a brief pulse of at least $\sim$25 m sea-level rise, based on the smoothed record (or up to $\sim$50 m based on the unsmoothed time series), occurred between $\sim$141 and 138 ka (identified as MWP-2A in Marino et al. (2015),

Fig. 7a). This pulse was followed by a slight sea level fall (∼10 m in the smoothed record) between ∼139 and 138 ka. Finally, a more significant pulse of ∼70 m in RSL rise (MWP-2B) is inferred between 135 and 130 ka. The period between the ephemeral pulse of sea-level rise at the beginning of the PDG (MWP-2A) and the second prolonged pulse (MWP-2B/HS11), has sometimes been referred to as the PDG sea-level reversal (Siddall et al., 2006).

The coral RSL data from Huon Peninsula and Tahiti independently provide additional evidence for an ephemeral reversal in sea level rise occurring during the penultimate deglaciation (Fig. 7). In the case of Tahiti, sedimentary evidence for the superposition of shallow and deeper water facies led to the interpretation that there was an ephemeral deepening (sea-level rise) followed by a return to shallower water conditions (sea-level fall or stabilization) (Thomas et al., 2009). The Tahiti data provide bounding ages on the timing of this sea level rise pulse, with ages of corals that grew at 135.0 ka (in 0-6 m water depth)
and 133.5 (±1) ka (0-25 m water depth). In between these shallower facies, there is a deeper water facies (≥20 m paleowater depth), but there are no reliable ages within this interval of the core (Thomas et al., 2009). This observation, based on changes in both the lithofacies and benthic foraminiferal assemblage, is interpreted as a pulse of sea-level rise in between about 135.0 and 133.5 ka (Fujita et al., 2010). A similar sea level oscillation has also been interpreted based on the stratigraphy as well as the age and paleowater depth reconstruction at Huon Peninsula (Esat et al., 1999). The absolute timing of coral growth is
only loosely constrained at this site due to open-system behaviour of the U-series isotopes (as reflected by the scatter in ages of corals collected in Aladdin's cave, ∼134 to 126 ka, Fig. 7) (Esat et al., 1999). Indeed, the corals from Terrace VII have ages (with high uncertainty) ranging from about 137 to 134.5 ka and the corals from the cave have a wide range of ages, from 134.1 to 125.9 ka (more details in Esat et al. (1999)). Given that the younger end of this age range is clearly within the MIS 5e sea-level highstand (e.g. Stirling et al., 1998), it is more likely that the older end of this diagenetic array of data from Aladdin's
cave is a better approximation for a the primary age (i.e. it is closer to the unaltered end member). Despite these diagenetic concerns, the agreement in the timing of this PDG sea level reversal (MWP-2A) in Tahiti and Huon Peninsula is striking (Fig. 7b).

    When considering the 95% probabilistic intervals of the Red Sea RSL reconstruction on the chronology from Grant et al. (2012), an overlap is observed with the coral data over the MWP-2A interval, within the stated uncertainties. Still, both coral
dataset suggest that MWP-2A occurs several millennia later (i.e. ∼135-134 ka) than in the Red Sea RSL reconstruction. This mismatch is likely to be related to the difficulty to precisely anchor the dating of the current Red Sea RSL age scale over this interval (as also discussed in the supplementary information of Grant et al. (2012)). Hence, we propose a revised chronology for the Red Sea RSL record in order to provide a better agreement with the absolutely-dated corals. Given the potential ambiguities of the tie point defined in Grant et al. (2012) to stretch the depth scale across this interval, we find it reasonable to adjust it such
that the timing of MWP-2A is more consistent with the absolute ages provided by the Tahiti and Huon Peninsula coral data.

    We note that reassigning the tie points across this interval (Table S5), where tie points are placed at the beginning and end of MWP-2A (as defined by the coral data), results in a sea-level reconstruction that more closely approximates a linear age-depth model (Fig. 7a). This revised age model for the Red Sea RSL is adopted as our preferred reconstruction for sea-level change during the PDG. This reconstruction also compresses the total duration of the sea-level rise during the entirety of the

PDG transition, which has implications for the freshwater forcing in the NH and for making analogies between the last and penultimate deglaciations.

This revised chronology is still attached to large uncertainties given the limits of the datasets. Also, considerable uncertainties remain with the magnitude of the sea-level pulse during MWP-2A because some of the corals cover a wide range of paleowater depth (0 to 6 m for the pre-MWP-2A Tahiti corals, ≥20 meters for the Tahiti corals during MWP-2A, 0-25 m for the post-MWP-2A Tahiti corals and 0 to 20 m for the Aladdin's Cave corals). Despite these uncertainties in the absolute position of sea level, the relative sea-level changes for each site clearly demonstrate an ephemeral deepening during meltwater pulse MWP-2A in both cases.

Glacial isostatic adjustment to the deterioration of the PGM ice-sheets will also differentially affect Tahiti and Huon Peninsula, which precludes a direct comparison of the magnitude of sea-level change between these sites or a direct interpretation of global mean sea-level change in the absence of modelling. Because the changes in global mean sea level are rapid across the penultimate deglaciation, the eustatic signal is likely dominant, leading to a timing of the rapid changes that is similar between local RSL and global mean sea-level reconstructions. Still, the rate of change may be different between sites due to local differences in the magnitude of sea-level change. Based on the revised chronology for the Red Sea RSL and on the coral constraints, MWP-2A starts at ∼137 ka, while MWP-2B starts at ∼133 ka (Fig. 7, Table 4).

Finally, far-field coral data from the Seychelles and Western Australia, that have been corrected for the glacial isostatic adjustment (e.g. Dutton et al., 2015), indicate that global mean sea level passed the position of modern sea level at about 129 ka (Fig. 7). The evolution of sea level during the LIG high-stand is still debated and may have included some meter-scale sea-level oscillations, but by at least some accounts, it is thought to have risen a few meters between 129 and 122 ka (e.g. Kopp et al., 2009; Dutton et al., 2015). So while the timing of peak sea level may have occurred later in the interglacial (∼122 ka), the onset of the highstand (∼129 ka) could represent an inflexion point in the rate of sea-level change coming out of the rapid deglaciation and into the interglacial.

Overall, eustatic sea-level reconstructions based on paleodata and continental ice-sheets simulations (Section 4) are consistent. However, the amplitude of the eustatic sea level change across the PDG estimated from the Red Sea reconstructions is ∼10 m smaller than the combined ice-sheets simulations (Fig. 4e). In addition, the sea-level data suggest a small sea-level increase at ∼140 ka, which is not present in the ice-sheet simulations. Both suggest that the main phase of sea-level rise/continental ice-sheets disintegration initiates at ∼134 ka, even if the overall magnitude is larger in the ice-sheet simulation, but with a lower rate of change than in the Red Sea reconstructions. However, it is worth keeping in mind that both the sea level data and ice-sheet model-based approaches are associated with large uncertainties regarding the exact timing and amplitude of global sea-level changes across the PDG.

## 6 Protocol for transient simulations of the PDG

To this date, no transient simulation covering the period 140 to 127 ka has been performed with a 3-dimensional climate model. However, the climate evolution over the period 135 to 115 ka has been successfully simulated with the Earth system

model LOVECLIM (Loutre et al., 2014; Goelzer et al., 2016a). Their simulations highlight the potential and feasibility of transient simulations of the PDG. In addition, as a proof of concept, transient experiments of the last deglaciation have been successfully performed with AOGCMs and Earth system models (Liu et al., 2009; Menviel et al., 2011; Roche et al., 2011; Gregoire et al., 2012; He et al., 2013; Otto-Bliesner et al., 2014). As detailed in the previous sections, the proposed transient simulation of the PDG will use boundary forcings consistent with the ones used for the last deglaciation (Ivanovic et al., 2016): i.e. appropriate orbital parameters, greenhouse gases concentration, continental ice-sheets geometry and meltwater input (Figure 1). To maximise the use of the transient simulations, the transient simulations of the penultimate and last deglaciations, as well as the *piControl* should be performed with the same version of the climate model.

## 6.1 Equilibrium spin-up at 140 ka (*PDGv1-PGMspin*)

If a LGM run is already available, then it is suggested to initialise the 140 ka spin-up from the climate fields and ocean state produced by the LGM equilibrium run. Starting from a LGM state may minimize the duration of the spin-up, as it should shorten the time to reach equilibrium in the deep ocean (Zhang et al., 2013). If starting from a pre-industrial set up, it is suggested to follow the PMIP4 LGM protocol (Kageyama et al., 2017) to set up the 140 ka state, but using the 140 ka boundary conditions described below instead of the respective 21 ka boundary conditions.

The model should be forced with 140 ka background conditions (Table 1), including appropriate orbital parameters, GHG concentrations as averaged over the interval 141-139 ka (191 ppm $CO_2$, 385 ppb $CH_4$, and 201 ppb $N_2O$), as well as the NH and Antarctic ice-sheets' extent, topography and associated albedo (as described above). The forcing file describing the evolution of NH and Antarctic ice-sheets, as simulated by ice-sheets models (Abe-Ouchi et al., 2013; Briggs et al., 2014) (Section 4), is available in the Design and Data sections of the PMIP4 wiki. They include the evolution of the ice-mask, as well as surface and bedrock elevations. Kageyama et al. (2017) provides guidelines for computing land-ice fraction and orography from the ice-sheet reconstruction datasets. The details of these forcings and the approach taken to compute them will ultimately depend on each model resolution.

The large glacial ice-sheets of the PGM impacted sea level and the land-sea mask. It would be best to modify the land-sea mask resulting from ice-sheet changes. Depending on the resolution of the model, this might not be a crucial parameter, except for some bathymetry and land-sea mask features, which have particular importance for ocean circulation. For example, the Bering Strait, which is 40 to 50 m deep, exerts a significant control on NADW and North Pacific Intermediate Water formation (e.g Okazaki et al., 2010; Hu et al., 2012). During the 140 ka spin-up, the Bering Strait should be closed. Following the recommendations for the PMIP4 LGM equilibrium run (Kageyama et al., 2017), the land-sea mask should include the exposure of the Sahul and Sunda shelves in the Indo-Australian region, as well as closure of the strait between the Mediterranean Sea and the Black Sea.

To account for the maximum PGM ice-sheet expansions, and associated global sea-level of ∼100 m below pre-industrial times, global salinity should be set at +0.85 psu above pre-industrial level. Furthermore, if oxygen isotopes are included in the simulations, the ocean mean $\delta^{18}O$ should be initialized at 1‰ and if a carbon cycle model is included, the global mean

alkalinity content should be increased by about 80 $\mu$mol/L. Compared to the LGM state, these values are 0.15 psu lower for global salinity, 0.2‰ lower for mean ocean $\delta^{18}$O and about 16 $\mu$mol/L lower for global alkalinity.

The model should be spun-up until near equilibrium is reached. Previous PMIP protocols recommend that the simulations are considered at equilibrium when the trend in globally averaged SST is less than 0.05°C per century and the Atlantic Merid-ional Overturning Circulation (AMOC) is stable. Marzocchi and Jansen (2017) recently pointed out that the AMOC should be monitored on a centennial timescale to properly assess equilibrium. Zhang et al. (2013) further suggested that the trend in zonal-mean salinity in the Southern Ocean (south of the winter sea-ice edge) should remain small (less than 0.005 psu per 100 years), especially in the Atlantic sector, to avoid potential transient characteristics in the deep ocean from impacting on AMOC strength. For models including representations of the carbon cycle or dynamic vegetation, the requirement is that the carbon uptake or release by the biosphere is less than 0.01 PgC per year. Similar to the recommendation made in Kageyama et al. (2017), the outputs of at least 100 years of the equilibrated 140 ka spin-up should be made available and fully described.

## 6.2   Transient forcings across the PDG (*PDGv1*)

The main changes in boundary conditions across the deglaciation, i.e. insolation, GHG concentrations and continental ice-sheets, have been described in sections 2, 3 and 4, respectively and are summarised in Table 1. For all simulations, methods should be fully documented.

### 6.2.1   Orography, bathymetry, coastlines and rivers

Disintegration of continental ice-sheets during the deglaciation affected continental topography and ocean bathymetry, and thus coastal outlines and river routing. Therefore, time-varying changes in land-ice fraction, land-sea fraction, and topography should be applied. Variations in the ice mask and topography should be updated at the same time. It is up to each group to decide the appropriate time frequency at which to update this forcing. The resolution of the files provided is 500 years, but higher frequency changes obtained through linear interpolations can also be performed to avoid step changes. As mentioned for the 140 ka spin-up regarding changes in land-sea fraction, particular attention should be given to the opening of the Bering Strait and the flooding of the Sunda and Sahul shelves. When possible, these should be varied across the PDG as ice-sheets disintegrate and sea-level rises. Following the combined ice-sheet history presented here, which includes some GIA adjustment, the Bering Strait might open at ~127.5 ka. Please note that this is a first estimate, which is associated with large uncertainties.

As changes in the land-sea mask could impact water delivery to the ocean through rivers, River mouths should be consistent with the adjusted land-sea mask (Kageyama et al., 2017). If possible and of interest, river networks could also be remapped to take into account the ice-sheet changes.

### 6.2.2   Vegetation, land surface and other forcings

The climatic and ice-sheet changes occurring between 140 and 127 ka will significantly impact the vegetation, and thus also land albedo, evapo-transpiration and the terrestrial carbon pool. The preferred option would thus be to include a dynamical

vegetation model, fully coupled to the atmospheric model. Care should be taken in regions where an ice-sheet is present, as the ice-sheet and its albedo should replace any possible vegetation. If a coupled dynamical vegetation model is not available, then the experiments should be run with prescribed land-surface parameters and with fixed vegetation types and plant physiology outside of the regions covered by a continental ice-sheet, as obtained from the CMIP5 pre-industrial set up (Taylor et al., 2012; Ivanovic et al., 2016). In that case, care should be taken to have a consistent land-surface/vegetation forcing with the adjusted land-sea mask. Vegetation/land-surface type on a newly emerged land (for example the Sunda shelf) will thus need to be fixed based on interpolations with the nearest grid points.

### 6.2.3 Freshwater forcing

Through their impact on global salinity and ocean circulation, disintegrating ice-sheets can significantly affect the climatic and biogeochemical evolution of the penultimate deglaciation (Oppo et al., 1997; Cheng et al., 2009; Hodell et al., 2009; Landais et al., 2013; Deaney et al., 2017). In particular, meltwater input in the North Atlantic region could be a significant driver of changes in NADW formation (Loutre et al., 2014; Goelzer et al., 2016b; Stone et al., 2016), including the ones associated with HS11 (further discussed in Section 8.2). It is thus strongly recommended to include a carefully designed freshwater scenario when performing transient simulations of the deglaciation.

As much as possible, and for all scenarios, meltwater should be added in the appropriate locations to match the evolution of the ice-sheets. Freshwater can be added over an appropriate ocean area close to the disintegrating ice-sheet, or a self-consistent paleo surface drainage forcing could also be implemented. This would involve using the provided downslope routing fields to route the water flux (fwf, cm/yr) from each grid cell:

$$fwf = (P - E)_{GCM} - (dH/dt)_{ice-sheet} * 0.91,$$ (1)

with P for precipitation (cm/yr), E for evaporation (cm/yr) and $(dH/dt)_{ice-sheet}$ (cm/yr, assuming an ice density of 0.91 g/cm$^3$) the change in ice-sheet thickness over time as described in the ice-sheet forcing files. Modellers may need to adjust the above to account for land surface model changes in water storage. In case of negative meltwater forcing, the artificial salt flux addition should be spread globally over the ocean.

Estimates of sea-level changes across the PDG suggest a global sea-level rise of ∼100 m (Grant et al., 2012, 2014) (Fig. 7) between 140 and 130 ka, due to the disintegration of NH and Antarctic ice-sheets. As the Antarctic contribution to sea level is estimated at ∼13 m (Fig. 4), this leaves a ∼87 m contribution from NH ice-sheets.

For the NH, three meltwater scenarios are proposed based *i)* on changes in NH ice-sheets, as described in section 4 (Fig. 4f, black, fIC) (Abe-Ouchi et al., 2013), *ii)* on global mean sea-level changes with our revised chronology (see section 5, Fig. 4f, blue, fSL) minus a linear Antarctic contribution of 13 m (Fig. 4d) (Briggs et al., 2014), and *iii)* as derived from North Atlantic and Norwegian Sea IRD records (see also section 8.2, Fig. 4f, red, fIRD).

Estimates of meltwater input to the North Atlantic based on changes in NH ice-sheets (Fig. 4b,e,f, black, fIC) (Abe-Ouchi et al., 2013) suggest a sustained (≥0.1 Sv) meltwater flux between ∼133 and 127 ka. This forcing closely follows changes in high

northern latitude summer insolation and would probably lead to a significant NADW weakening during HS11, lasting until about 125.5 ka depending on the model's sensitivity.

Meltwater input estimates derived from Red Sea sea-level records (Fig. 4f, blue, fSL) on the revised chronology (section 5) display a large (up to 0.3 Sv) 1000-year-long meltwater pulse centred at 137 ka, a broad meltwater input between 134 and

131 ka with a large peak at 131.7 ka, and two relatively late meltwater pulses centred at 130 ka and 128.3 ka, respectively. The magnitude and length of the AMOC perturbation resulting from such a meltwater scenario will, of course, depend on each model's sensitivity and on the initial AMOC state. However, for most models, it is expected that NADW formation would weaken significantly between ∼137.8 and 136.5 ka as well as between ∼133.5 and 129 ka. Another small AMOC perturbation is expected between ∼129 and 127.8 ka in this scenario.

There are significant uncertainties associated with both the simulation of NH ice-sheets and the timing and amplitude of sea-level changes. In addition, to fully explore the potential of transient deglacial simulations, it is critical to simulate NADW changes in global agreement with proxy records. Since periods of increased IRD delivery have been associated with changes in NADW (e.g. van Kreveld et al., 2000; Rodrigues et al., 2017), we design an additional meltwater scenario, for which the timing is based on North Atlantic and Norwegian Sea IRD records (Fig. 8a). To construct the fIRD scenario, the IRD record of

MD95-2010 (Risebrobakken et al., 2007) is assumed to represent freshwater input only originating from the Eurasian ice-sheet for the period 140 to 133.8 ka, while the stack of IRD records presented in Figure 8b (black) is assumed to represent freshwater originating from both the North American and Eurasian ice-sheets for the period 133.7 to 127 ka. The normalized MD95-2010 IRD record and the IRD stack were thus scaled so as to obtain a total NH sle contribution of 87 m (Fig. 4e, red), with 35 m originating from the Eurasian ice-sheet and 51 m from the North American one. It is expected this scenario will lead to a

weakening of NADW between 139.5 and 136.5 ka as well as at ∼135 ka. The sustained meltwater pulse might induce NADW cessation between ∼133.5 and 129.4 ka, followed by a recovery sometime between 129 and 128 ka.

As can be seen in Figure 4f, the meltwater forcing scenarios based on sea-level changes (fSL) and the IRD record (fIRD) share some similarities. The main differences between these scenarios are the small meltwater pulse at ∼137 ka in fSL, which is of much reduced amplitude in fIRD, and the ∼128 ka pulse in fSL not present in fIRD. Finally, the fSL scenario includes

periods of significant negative meltwater forcing (i.e. artificial salt flux addition), corresponding to phases of sea-level lowering. As described above, it is suggested to use the self-consistent drainage scheme for the location of the meltwater input. For NH scenarios fSL and fIRD, $(dH/dt)_{ice-sheet}$ (from equation 1) would need to be scaled to obtain the appropriate meltwater flux.

For those who wish to take part in an inter-comparison involving comparable boundary conditions, the fSL scenario is put forward as the recommended option (Table 1). Depending on the sensitivity of the model to freshwater input, fIRD scenario

can provide a good alternative to fSL. However, the ultimate choice of the appropriate freshwater scenario is left to each group, and sensitivity experiments to assess the climatic impact of different meltwater scenarios are encouraged (section 7).

To take into account the effect of Antarctic ice-sheet melting, freshwater should also be added close to the Antarctic coast, following the self-consistent routing scheme described above (Fig. 4d) (Briggs et al., 2014). This scenario will broadly consist in adding ∼0.0135 Sv meltwater from 140 to 130 ka. However, there are significant uncertainties associated with the timing

of the Antarctic deglaciation. Additional experiments are necessary to further constrain the impact of the Antarctic ice-sheet

disintegration on the deglacial climate and carbon cycle (e.g. Menviel et al., 2010). Therefore, another scenario inspired by the Antarctic ice-sheet deglacial trajectory described in Goelzer et al. (2016a) is proposed (Table 2, fSL2).

Finally, if preferred and instead of the Northern and Southern Hemisphere freshwater scenarios described here, a globally uniform freshwater flux that corresponds to the ice-sheets evolution can be added to simply conserve salinity throughout the transient deglacial experiment (Table 2, fUN). This latter option is equivalent to the *melt-uniform* scenario used in the PMIP4 transient simulations of the last deglaciation (Ivanovic et al., 2016).

### 6.3 Requested variables

A transient simulation of the PDG implies that the climate model will need to be integrated for 13,000 years, thus generating a large amount of data. Depending on the spatial resolution of the different model components, storage of data outputs can become an issue. We therefore suggest a list of variables that absolutely needs to be uploaded to the ESGF database (Balaji et al., 2018) to understand the results and perform inter-model and model-data comparisons. The CMIP6 conventions should be used for all the variable names and units. Table 3 presents a list of forcings, and variables related to the atmosphere, ocean, vegetation and ocean biogeochemistry realms (if relevant) to be saved either as annual or monthly means. In addition to this list standard global, hemispheric or zonally averaged variables would need to be included, such as northern and southern sea-ice extent and surface air temperature, maximum meridional overturning streamfunction in the North Atlantic, top of the atmosphere (TOA) energy budget, surface energy budget, carbon budget (if relevant), etc... The list detailed in Table 3 represents the strict minimum of variables to be distributed. It is expected that additional variables will be included by each group, particularly if other model components are included (e.g. isotopes).

### 7 Sensitivity experiments

The penultimate deglaciation is a particularly interesting period as it provides a framework to study the impact of changes in insolation, atmospheric GHG concentrations and continental ice-sheets on climate. In addition, meltwater input associated with the disintegration of continental ice-sheets will also impact the oceanic circulation and thus global climate and biogeochemistry (e.g. Liu et al., 2009; Menviel et al., 2011, 2014; Schmittner and Lund, 2015; Goelzer et al., 2016a; Ivanovic et al., 2017; Menviel et al., 2018). Ultimately, transient deglacial simulations will inform on the impact of each of these processes as well as their interactions.

However, the transient simulation of the PDG proposed here might present a challenge to state-of-the-art Earth-system models, as they will need to include a spin-up with 140-ka boundary conditions and be run for 13,000 years. For this reason (i.e. to avoid the necessity for additional simulations when the computational expense is prohibitive), the main experiment includes all appropriate boundary forcing as well as meltwater input due to disintegrating ice-sheets. This experiment will provide valuable information on processes occurring during the PDG and will allow for a thorough comparison of the penultimate and the last deglaciation by complementing existing transient simulations of TI (Liu et al., 2009; Menviel et al., 2011; Roche et al., 2011; Gregoire et al., 2012; He et al., 2013) and the new experiments performed as part of PMIP4 (Ivanovic et al., 2016).

These transient simulations of the PDG will also complement existing transient simulations of the PDG performed with the LOVECLIM Earth-system model and covering the time interval 135 to 115 ka (Loutre et al., 2014; Goelzer et al., 2016a). Finally, the proposed experiment will provide a link to the PMIP4 transient simulation of the LIG (127 to 121 ka) as well as the PMIP4 127 ka timeslice experiment (*lig127k*) (Otto-Bliesner et al., 2017), even though the protocol of the LIG experiments

includes pre-industrial continental ice-sheets.

The main experiment is described in Table 1 and includes a comprehensive meltwater scenario. Additional transient simulations of the PDG are encouraged to, for example, assess different timing and amplitude of meltwater-input (Table 2), but also simulations with globally uniform meltwater input (fUN). As there are large uncertainties associated with meltwater input scenarios and the sensitivity of deep convection to the freshwater input, it is strongly advised to perform both the main experiment

and fUN to isolate the impact of the freshwater input (Goelzer et al., 2016a). In addition, the response to individual forcings (i.e. orbital parameters, GHGs and ice-sheet extent and albedo) could be assessed separately (He et al., 2013; Gregoire et al., 2015).

Even though there are some geological constraints on glacial evolution (see Section 4.1.), there remain large uncertainties associated with the reconstruction of continental ice-sheets during the PGM, across the PDG and during the LIG. In addition,

there are significant uncertainties associated with the parametrization of dynamical processes governing continental ice-sheets, and most importantly those representing the climate forcing. Through their impact on albedo and topography, continental ice-sheets can significantly influence climate (e.g. Timm et al., 2010; Zhang et al., 2014), even when the ice-sheets are small (e.g. Löfverström et al., 2014; Gong et al., 2015; Roberts and Valdes, 2017; Gregoire et al., 2018). Therefore, the impact of ice-sheet extent and topography on the PGM, across the PDG and during the LIG, should be studied in detail through sensitivity

experiments.

Models that include prognostic aerosol and/or dust could provide very useful fields to the community. Similarly, evaluating the impact of varying deglacial dust levels on climate and biogeochemistry are of crucial importance (Evan et al., 2009; Martínez-García et al., 2014). Therefore, sensitivity simulations forced with different dust-flux scenarios are encouraged.

## 8   Comparing model simulations to paleoclimate and paleoenvironmental reconstructions

It is central to PMIP to evaluate the realism of the coordinated transient simulations and the associated sensitivity experiments with environmental and climate reconstructions from archives. In the following, we first report on the few existing surface temperature syntheses that cover at least part of our interval of interest. We then provide a non-exhaustive selection of additional paleoclimatic and paleoenvironmental records extending back to 140 ka. This will allow a first-order comparison between the changes recorded in different parts of the climate system and those inferred in the transient simulations (Figs. 8 and 9,

Tables 4 and 5). The selection criteria are related to each record's temporal resolution and to the climatic or environmental interpretation that can be inferred from the measured tracer. We put a special emphasis on selecting marine sediment core records that can inform on millennial-scale changes occurring in the North Atlantic sector during HS11, potentially linked

with changes in NADW formation. This section ends with a review of the main limitations that will be associated with the model-data comparison and recommendations.

## 8.1 Available surface temperature syntheses

Quantitative comparisons between paleo-reconstructions and model outputs across the time interval 140-127 ka are possible, but remain limited to a few parameters (e.g. surface-air and sea-surface temperature (SAT and SST)). Qualitative and indirect comparisons are also adequate to evaluate simulations: for example, simulated AMOC strength against paleo-records (section 8.2), simulated precipitation compared to Chinese speleothem calcite $\delta^{18}O$ records or the simulated vegetation patterns against pollen-based vegetation reconstructions (section 8.3, Table 5).

There are currently no paleoclimate data compilations focusing on the PGM or covering the full length of the PDG, but syntheses of quantitative and temporal surface temperature changes focusing on the LIG are available for model evaluation of the second part of the penultimate deglaciation. Indeed, the synthesis by Capron et al. (2014) covers the time interval 135-110 ka and compiles, in a coherent temporal framework underpinned by the AICC2012 ice-core chronology, annual SAT above Greenland and Antarctica and summer SST records from the North Atlantic and the Southern Ocean located at latitudes above 40°. In Uemura et al. (2018), updated surface air temperature reconstructions displayed on AICC2012 are presented for the EDC, Dome F and Vostok deep ice cores. They represent useful constraints, particularly to investigate spatial differences in temperature between East Antarctica inland ice core sites during the PDG.

Further, based on harmonized chronologies between marine records, Hoffman et al. (2017) propose a global annual SST compilation with timeseries encompassing 130 to 115 ka. Hoffman et al. (2017) use the SpeleoAge age scale as a reference chronology. This SpeleoAge age scale results from the adjustment of the EDC3 ice-core timescale using radiometric dates from Chinese speleothems (Barker et al., 2011) and presents age differences across the PDG with AICC2012 (e.g. ~1.4 ka at 127 ka, ~1.0 ka at 136 ka). Both syntheses are useful to perform site-to-site model-data comparisons in order to provide detailed information about the spatial structure of the changes. The regional stacks produced by Hoffman et al. (2017) provide a first order estimate of the mean annual SST responses between 130 and 127 ka.

## 8.2 North Atlantic records to inform on Heinrich stadial 11 and NADW changes

Deglaciation of NH ice-sheets led to increased meltwater input into the North Atlantic, thus reducing surface water density and potentially weakening NADW formation. It has been shown that changes in NADW have a significant impact on North Atlantic and European climate (e.g. Stouffer et al., 2007; Martrat et al., 2014; Wilson et al., 2015), but also have a strong imprint on tropical hydrology (e.g. Arz et al., 1998; Chiang and Bitz, 2005; Deplazes et al., 2013; Otto-Bliesner et al., 2014; Jacobel et al., 2016). Through oceanic and atmospheric teleconnections, NADW variations can further modulate the strength of the Asian Monsoon (e.g. Cheng et al., 2009) and the formation of other oceanic water masses, such as North Pacific Intermediate Water (Okazaki et al., 2010). Using a multi-proxy approach, we investigate here the millennial-scale variability occurring during the PDG, and particularly potential changes in NADW linked with the meltwater input of HS11.

We selected 11 marine sediment cores from the Western Mediterranean Sea, the North Atlantic and the Nordic Seas with relatively high-resolution proxy data across the PDG (Tables 4 and 5, Fig. 8). These sites have undergone multi-proxy investigations (e.g. stable isotopes on benthic and planktonic foraminifera, foraminifera faunal assemblages, IRD counts, SST reconstructions, Pa/Th, $\epsilon_{Nd}$). The chronology of all the sites has been revised here using the radiometrically-dated time scale of the Italian Corchia cave (original record in Drysdale et al. (2009), see also Tzedakis et al. (2018) for the high-resolution calcite $\delta^{18}O$ profile) as a reference for the PDG (Text S1 and S2, Table S1).

Phases of meltwater input, and/or changes in NADW can be investigated using different proxies. The intensity of NADW formation can be deduced from variations in Pa/Th (a kinematic proxy for the rate of deep water export from the North Atlantic) and $\epsilon_{Nd}$ (a water provenance tracer) in North Atlantic sediment cores (McManus et al., 2004; Roberts et al., 2010), but only one record (ODP 1063) currently covers the PDG and the time-resolution of Pa/Th and $\epsilon_{Nd}$ measurements across the PDG is relatively low (Böhm et al., 2015; Deaney et al., 2017). Phases of iceberg discharges, which might significantly impact NADW formation, can be inferred from the amount of IRD in North Atlantic cores. For this time period, IRD records with sufficient resolution are available from six Atlantic cores (CH69-K09, ODP980, ODP983, ODP984, SU90-03 and ODP1063) and one Norwegian Sea core (MD95-2010). The IRD records were first normalized, after which an IRD stack was made for the North Atlantic cores by interpolating them onto a common age-scale with a regular (100-year) time-step.

Because continental ice-sheets are $^{18}O$-depleted, the meltwater supply also induces a drop in seawater $\delta^{18}O$ ($\delta^{18}Osw$), particularly at the ocean's surface. For six of the sites, changes in $\delta^{18}Osw$ have been reconstructed from paired SST and planktic $\delta^{18}Oc$ records using the following paleo-temperature equation (Shackleton, 1974):

$$Tiso = 16.9 - 4.38 * (\delta^{18}Oc + 0.27 - \delta^{18}Osw) + 0.10 * (\delta^{18}Oc + 0.27 - \delta^{18}Osw)^2 \qquad (2)$$

with *Tiso* the isotopic or calcification temperature (°C) that is deduced from SST reconstructions; $\delta^{18}Oc$, the isotopic composition of the calcite (‰, PDB); $\delta^{18}Osw$, the isotopic composition of seawater (‰, SMOW). The factor 0.27 is added for calibration against international standards (SMOW vs. VPDB) (Hut, 1987).

As a weakened NADW transport leads to the accumulation of remineralized carbon below 2500 m in the North Atlantic (Marchal et al., 1998), changes in oceanic circulation are also inferred from benthic foraminifera $\delta^{13}C$, a tracer for the ventilation of deep-water masses (Duplessy et al., 1988; Curry et al., 1988; Menviel et al., 2015; Schmittner et al., 2017). Finally, since a NADW weakening reduces the meridional heat transport to the North Atlantic (Stouffer et al., 2007), it is generally accompanied by a sea surface cooling across this region (Kageyama et al., 2013; Ritz et al., 2013).

The Norwegian Sea core MD95-2010 (Risebrobakken et al., 2006) indicates an increased IRD content starting at 139.5 ka (Fig. 8b). This could suggest an initiation of the penultimate deglaciation by enhanced iceberg calving (and subsequent melting) from the Fennoscandian ice-sheet. However, this has little to no effect in the North Atlantic. Only in the western side of the North Atlantic, right in the meanders of the Gulf Stream, does core CH69-K09 display a surface cooling and a drop in $\delta^{18}Osw$ starting at ~137.5 ka (Labeyrie et al., 1999).

A more significant deglacial pulse, indicated by an IRD increase in both the Nordic Seas and the North Atlantic, as well as a surface freshening in all North Atlantic cores occurs at ~136.4-133.9 ka (Fig. 8g). This meltwater input, which could

correspond to the first phase of HS11, could have led to a significant weakening of NADW, as indicated by $\epsilon_{Nd}$ and Pa/Th records from the Bermuda rise (Böhm et al., 2015) (not shown), as well as by the accompanying surface cooling in the North Atlantic (Chapman and Shackleton, 1998; Martrat et al., 2007, 2014; Deaney et al., 2017) and a decrease in benthic $\delta^{13}$C below 2000 m depth (Labeyrie et al., 1999; Shackleton et al., 2003; Hodell et al., 2008; Deaney et al., 2017). The low-resolution $\epsilon_{Nd}$

and Pa/Th records from the Bermuda rise (Böhm et al., 2015) also indicate a weakening of NADW transport during this period (not shown). This deglacial phase identified in North Atlantic marine sediment cores is broadly coincident with MWP-2A (Fig. 7, section 5).

Martrat et al. (2014) identified a double-u structure in SST records from the western Mediterranean Sea (ODP976) and the Portuguese margin (MD95-2042, MD01-2444) during the PDG, with a short-duration warming centred at ∼133.6 ka. This

warming is also seen in other North Atlantic records, such as ODP 984, 983, 980 and 1063 (Fig. 8 c-f). Very interestingly, and within dating uncertainties, this event corresponds to a small increase in atmospheric $CH_4$ (Figs. 1 and 2) (Landais et al., 2013), and broadly corresponds to a pause in the sea-level rise. In parallel, Pa/Th in core ODP1063 displays a significant decrease at this time (not shown), and benthic $\delta^{13}$C increases in the deepest North Atlantic cores (Fig. 8). This warming could thus be due to a short lived reinvigoration of NADW formation.

In agreement with previous studies (e.g. Marino et al., 2015; Tzedakis et al., 2018), the main phase of HS11 occurs between ∼133.3 and 130.4 ka and is characterised by a large IRD peak, cold surface-water conditions, a minimum in seawater $\delta^{18}$O , and low benthic $\delta^{13}$C values in most sites of the North Atlantic (Fig. 8). Elevated rates of iceberg calving and melting of NH ice-sheets thus lead to a large meltwater pulse in the North Atlantic, a significant global seal level rise (MWP2-B) (Grant et al., 2012, 2014), NADW weakening (Böhm et al., 2015) and a surface cooling that is at least regional.

The resumption of NADW formation at the end of HS11 marks the end of the penultimate deglaciation (Tzedakis et al., 2012) (∼130.4-128.5 ka). It is characterised by an increase in SST, seawater $\delta^{18}$O (Oppo et al., 2006; Mokeddem and McManus, 2016; Martrat et al., 2007, 2014; Deaney et al., 2017) and benthic $\delta^{13}$C from the North Atlantic region (Fig. 8). The associated atmospheric warming in the North Atlantic region also leads to terrestrial regrowth and thus a sharp increase in atmospheric $CH_4$ occurring between ∼129 and 128.5 ka (Landais et al., 2013).

**8.3  Other environmental and climate reconstructions**

Figure 9 shows a non-exhaustive selection of other environmental and climate proxy records. Tables 4 and 5 further provide the climate or environmental interpretation for each selected tracer to aid the model-data comparison, including the tracers described in section 8.2. Only a few speleothem calcite $\delta^{18}$O ($\delta^{18}$Oc) records and their environmental or climatic interpretation are presented here. However, it can be complemented with the compilation of speleothem $\delta^{18}$Oc records covering the time

interval 140 to 110 ka presented by Govin et al. (2015). It is also important to recognise that the dominant drivers of speleothem $\delta^{18}$Oc may change over time and differ from one cave site to another (see Section 8.4). Therefore we strongly advise a critical assessment of these interpretations based on the most recent developments and advances in stable-isotope hydrology and the original publications.

Tables 4 and 5 also detail the timing of the major changes (and associated uncertainties) as recorded in each timeseries. These major changes are identified using the RAMPFIT or BREAKFIT software programs (Mudelsee, 2000, 2009). Age uncertainties ($1\sigma$) reported in Tables 4 and 5 include *i)* the "internal" error of the event given by RAMPFIT or BREAKFIT, *ii)* the relative error related to the climatic alignment method for marine sediment and Ioannina records and *iii)* the absolute dating error of the reference time scale. The selected ice- and sediment-based records are all displayed on AICC2012 or on a timescale coherent to this ice-core age scale, while terrestrial records are displayed on their own independent chronology. Among these paleoclimatic time series, several isotopic records are presented here and we strongly encourage transient simulations to be performed with oxygen and/or carbon isotope-enabled models to allow direct quantitative comparison between simulated and measured isotopic time series.

Little change occurs until the beginning of phase 1 at ~136.4 ka, after which a cooling phase is identified in a few records of the North Atlantic (Fig. 8 and Fig. 9a, major change [A]). This also corresponds to reduced monsoon activity as recorded in Chinese speleothems (Fig. 9d [M]), and the initiation of the Antarctic warming (Fig. 9h [W]). The short-lived warming event in the North Atlantic associated with phase 2 (Fig. 9a [B]) is also identified in the Chinese speleothems as a slightly wetter interval (Fig. 9d [N]). Other environmental records might not have the necessary resolution to record this multi-centennial-scale event. The main phase of HS11, corresponding to phase 3, is associated with meltwater input and cold conditions in the North Atlantic (Fig. 9a [C] and 9c [I]), dry conditions over Europe (Fig. 9b) and Asia (Fig. 9d, interval between [O] and [P]), and warmer conditions at high southern latitudes (Fig. 9f, h). The end of HS11 (phase 4) associated with a pause in the meltwater input (Fig. 9c [J]) and progressively warmer conditions in the North Atlantic and southern Europe (Fig. 9a [D, F] and 9c [K]) corresponds to a strengthening of the Asian monsoon (Fig. 9d [P] and 9e [Q]), and maximum warmth at high southern latitudes (Fig. 9f [R, S] and 9h [X]). Interglacial conditions in atmospheric $CO_2$ and $CH_4$ as well as North Atlantic temperatures and ventilation are attained at about 128.5 ka (Fig. 8), which is also associated with warm and wet conditions in southern Europe (Fig. 9b [G], and 9c [L]).

## 8.4 Limitations and recommendations

One important consideration to account for when comparing simulated variables against paleodata between 140 and 127 ka, is the large uncertainties associated with both absolute and relative chronologies of most paleoclimatic records during this time interval. Dating uncertainties range from a few centuries to up to several thousand years depending on the type of archive and dating methods (Govin et al., 2015). For instance, the average absolute dating error attached to the Corchia Cave $\delta^{18}O$ record is ~0.7 ka ($2\sigma$) (Tzedakis et al., 2018). For marine sediment chronologies, which are mainly based on record alignment strategies (e.g. a record on a depth scale is aligned onto a dated reference record), the age uncertainties encompass a relative dating error ("alignment error") in addition to the absolute error attached to the chronology of the dated reference record. As a result, the overall $2\sigma$ age error associated with North Atlantic sediment core ODP976 aligned onto the Corchia record is 1.6 ka on average (Table S2). Between 140 and 127 ka, the average absolute error attached to the AICC2012 chronology used to display ice and gas records from the EDC ice core is about 4 ka ($2\sigma$) (Bazin et al., 2013; Veres et al., 2013). This large AICC2012 dating uncertainty is thus attached to the GHG concentration records used to force the transient simulations (Section

3). It will therefore taint the relative timing of the changes in orbital and atmospheric $CO_2$ forcing that will largely drive the simulated evolution of climate and environmental changes across the PDG. The relative timing of those changes will also be affected by uncertainties attached to the temporal evolution of the ice-sheets and related meltwater forcing scenarios. However, uncertainties attached to the relative timing of changes between GHG forcing and the simulated changes in Antarctica and

the Atlantic Ocean basin are somewhat reduced since the ice- and sediment-based records are also displayed on AICC2012 timescale or on time scales coherent to the reference ice-core age scale (Section 8.2).

Limitations are also attached to the potential misinterpretation of climate and environmental proxies due to incomplete understanding of how some of those archives record climatic and environmental change. First, SST records highlighted here have been reconstructed using various microfossil and geochemical methods. Although the use of various tracers is known to

yield SST discrepancies in particular above 35°N (MARGO project members, 2009), the extent to which these different SST reconstruction methods influence the representation of temporal climatic changes across our studied time interval is poorly known. Additional difficulties arise from the individual methods commonly used to reconstruct past SST due to, for example, the poor understanding of the modern habitat (e.g. living season and water-depth) of microfossil species (e.g. foraminifera) (Jonkers and Kučera, 2015) and alkenone producers (e.g. Rosell-Melé and Prahl, 2013). With these limitations acknowledged,

model-data comparisons should use annual or appropriate seasonal climatic variables depending on the interpretation of the measured climate proxy proposed in the original or subsequent publications.

Regarding SAT reconstructions over Antarctica, the impact of changes in seasonality of snow precipitation on the reconstruction remains difficult to quantify based on ice-core water-isotopic records (e.g. Masson-Delmotte et al., 2011; Uemura et al., 2012). In addition, Sime et al. (2009) suggested that the temperature at Dome C should be much higher than the one inferred

from water isotopes and a constant temperature/$\delta^{18}O$ slope during MIS 5e. For now, we suggest the use of simulated annual mean climatic variables for the comparison. However, it is crucial that the seasonality of paleo-records is better assessed to improve the interpretation of temperature reconstructions, and hence the model-data comparisons.

In addition, uncertainties remain regarding the dominant controlling factors (i.e. changes in temperature, rainfall amount and rain sources) of speleothem calcite isotopic records (e.g. Govin et al., 2015). For instance, $\delta^{18}Oc$ records throughout Asia

are commonly interpreted as tracers of past changes in the intensity of the Asian monsoon. In particular, Chinese speleothem $\delta^{18}Oc$ is classically interpreted as reflecting the East Asia monsoon (e.g. Cheng et al., 2009, 2016). However, a water-hosing experiment performed with an oxygen-isotopes-enabled model suggests instead that $\delta^{18}Oc$ variations may reflect changes in the intensity of the Indian, rather than East Asian, monsoon precipitation during Heinrich events (Pausata et al., 2011; Caley et al., 2014). Another recent model study also demonstrates that variations in ice core and speleothem oxygen isotope

reconstructions cannot solely be attributed to climatic effects, but also reflect depleted $\delta^{18}O$ of nearby oceans during glacial meltwater events (Zhu et al., 2017). Overall, we strongly encourage the use of isotope-enabled models to allow for direct and quantitative model-data comparison of isotopic tracers.

Additional multi-centennial-scale paleoclimatic reconstructions are necessary in order to further constrain the millennial scale variability during the PDG. It is also crucial that comprehensive data compilation work is carried out over the entire

studied time interval and covering, in particular, the PGM in order to test the robustness of the initial 140 ka spin-up climate.

Modelling groups running transient deglacial simulations, and/or associated sensitivity experiments, are encouraged to use multiple paleorecords for a full diagnosis of the simulations.

## 9   Conclusions

Here, we present a protocol for performing transient simulations of the PDG spanning 140 to 127 ka. Changes in boundary
conditions across the PDG that will serve as a forcing are presented and discussed. This includes changes in orbital parameters, GHG concentration, NH and Antarctic ice-sheets, and associated deglacial meltwater input. While not used as a direct forcing, changes in global sea-level are also presented on a new chronology. Finally, a series of key paleoclimatic and paleoenvironmental records are suggested to perform model-data comparisons. The marine records were recovered from the North Atlantic and Southern Ocean, while the continental records were retrieved from Europe, China and Antarctica. Performing transient
simulations with oxygen- and/or carbon-isotope-enabled Earth-system models could significantly improve model-data comparisons by providing a more direct and quantitative comparison with paleoproxies based on measured isotopic signatures (e.g. $\delta^{18}$O, $\delta^{13}$C).

Simulations of the penultimate deglaciation would allow a comparison with the last deglaciation, therefore highlighting similarities and differences between the last two deglaciations. The evolution of insolation across the two deglaciations is
different, potentially explaining the relatively more rapid disintegration of NH ice-sheets during the PDG compared to the last deglaciation. Acting both as a response to and driver of changes, atmospheric $CO_2$ appears to increase much more gradually during the penultimate than the last deglaciation. Another striking difference between the penultimate and the last deglaciation is the lack of a Bølling-Allerød warming in the middle of the deglaciation, even though, as discussed in sections 5 and 8, several records show a brief reversal centred at about 133.6 ka. Transient simulations can thus shed light onto the different
and interactive roles of radiative forcing (insolation, GHG concentrations) and ocean circulation changes (e.g. NADW and Southern Ocean ventilation) in driving climate change across the penultimate and the last deglaciations. Transient simulations performed with Earth-system models that include a dynamic vegetation and a global carbon cycle model, would be particularly useful in assessing the impact of climate change on vegetation cover and on marine ecosystems. The ultimate goal would be to perform transient simulations of the penultimate deglaciation with Earth-system models that include interactive ice-sheets and
carbon cycle components. But models might not be quite ready for such a task yet.

Transient simulations of the PDG and associated model/paleo-proxy comparisons provide a great opportunity to understand drivers and processes involved in one of the largest natural global warming period of the Quaternary. In addition, these transient simulations of the PDG will provide a bridge with the proposed PMIP4 transient simulations of the LIG (127 to 121 ka), and the 127-ka time-slice experiment (*lig127k*) (Otto-Bliesner et al., 2017). The PDG is a key period to understand as it led to the
LIG, an interglacial displaying warmer conditions than pre-industrial (e.g. Hoffman et al., 2017) as well as a global sea-level 6 to 9 m higher than today (e.g. Dutton et al., 2015).

*Data availability.* The combined ice-sheet and meltwater scenarios are available in the Supplement. The GHG data can be found at https://doi.org/10.1594/PANGAEA.871273. The combined ice-sheet, as well as the separate Northern Hemispheric and Greenland ice-sheets are also publicly available on the Research Data Australia repository at http://handle.unsw.edu.au/1959.4/resource/collection/resdatac_874/1 and https://doi.org/10.26190/5d0c0c0bd1f26.

In addition, all the forcing files as well as the paleo-data described in the manuscript are available on the PMIP4 wiki: https://pmip4.lsce.ipsl.fr/doku.php/exp_design:degla_t2.

*Author contributions:* LM and EC led and coordinated the study, and wrote sections 1 and 9. BOB wrote section 2. EC wrote section 3 in collaboration with AL, IO, KK and EW. LT and AA provided the continental ice-sheets data. LM wrote section 4 in collaboration with LT, AA, PCT and PLG. AD reassessed the timing of the sea-level changes across the PDG and wrote

section 5 in collaboration with EC. LM wrote section 6 and 7 in collaboration with BOB, RFI, LG, FH, MK, LT, and XZ. AG and EC in collaboration with RND reassessed all the chronologies of the paleo-records presented in the study. AG, EC, LM and RDN wrote section 8. All the authors designed the goals of the study, and contributed to the different parts of the manuscript.

*Acknowledgements.* This study was performed as part of the PAGES-PMIP working group on Quaternary Interglacials (QUIGS) and was initiated during the second QUIGS workshop at the Université du Quebec in Montreal, 18-20 October 2016. PMIP is endorsed by the

World Climate Research Program (WCRP) and CLIVAR. We thank J.-Y. Peterschmitt for setting up the the PDG page on the PMIP4 wiki. LM acknowledges funding from the Australian Research Council grants DE150100107 and DP180100048. EC received funding from the European Union's Seventh Framework Programme for research and innovation under the Marie Skłodowska-Curie grant agreement nb 600207. EW is supported by a Royal Society professorship. His part in this project has received funding from the European Research Council (ERC) under the European Union's Horizon 2020 research and innovation programme (grant agreement nb 742224). PCT and RFI

acknowledge funding from from NERC (NE/G00756X/1 to PCT and NE/K008536/1 to RFI). MK is funded by the CNRS. AD acknowledges U.S. National Science Foundation (NSF) grants 1559040 and 1702740, as well as the PALSEA working group. BOB's contributions are based upon work supported by the National Center for Atmospheric Research, which is a major facility sponsored by the U.S. National Science Foundation under Cooperative Agreement No. 1852977, with additional funding provided by an NSF P2C2 grant (AGS-1401803). F.H. was supported by the NSF (AGS-1502990) and by the NOAA Climate and Global Change Postdoctoral Fellowship program, administered by

the University Corporation for Atmospheric Research. XZ is supported by Helmholtz Postdoc Program (PD-301) and the Chinese "The Thousand Talents Plan" Program. AA, KK and IO acknowledge support by JSPS KAKENHI grants (17H06104 to AA, 15KK0027 and 26241011 to KK, 17K12816 and 17J00769 to IO), and MEXT KAKENHI grant (17H06323 to AA and 17H06320 to KK). The manuscript has benefited from helpful comments from two anonymous Reviewers as well as from the Editor Julia Hargreaves.

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

| Forcing | 140 ka spin-up | Transient simulation (140-127 ka) |
|---|---|---|
| | *PDGv1-PGMspin* | *PDGv1* |
| **Orbital parameters** | | |
| Eccentricity | 0.033 | from Berger (1978) |
| Obliquity | 23.414° | from Berger (1978) |
| Perihelion - 180° | 73° | from Berger (1978) |
| **Atmospheric greenhouse gases concentrations** | | |
| on the AICC2012 chronology (Bazin et al., 2013; Veres et al., 2013) | | |
| $CO_2$ | 191 ppm | Spline from Köhler et al. (2017) based on the EDC records |
| | | (Lourantou et al., 2010; Schneider et al., 2013) |
| $CH_4$ | 385 ppb | Spline from Köhler et al. (2017) based on the EDC record |
| | | (Loulergue et al., 2008) |
| $N_2O$ | 201 ppb | Linear increase from 201 ppb at 140 ka to 218.74 ppb at 134.5 ka |
| | | Spline from Köhler et al. (2017) based on the EDC and TALDICE records |
| | | (Schilt et al., 2010; Spahni et al., 2005) |
| **Ice-sheets*** | | |
| North American and Eurasian | 140 ka | IcIES-NH (Abe-Ouchi et al., 2013) |
| Greenland | 140 ka GSM-G | GSM-G (Tarasov et al., 2012) |
| Antarctica | 140 ka GSM-A | GSM-A (Briggs et al., 2014) |
| **Bathymetry and orography** | | |
| Bering Strait | closed | gradual opening consistent with sea-level rise |
| Sunda and Sahul shelves | emerged | gradual flooding consistent with sea-level rise |
| **Freshwater input*** | | |
| Northern Hemisphere | none | based on sea-level changes (fSL, blue in Fig. 4f) |
| Antarctic coast | none | 0.0135 Sv between 140 and 130 ka (constant rate) |

**Table 1.** Summary of forcings and boundary conditions to apply for the PGM spin-up (140 ka) and subsequent transient simulation of the PDG.

Yau, A., Bender, M., Robinson, A., and Brook, E.: Reconstructing the last interglacial at Summit, Greenland: Insights from GISP2, Proceedings of the National Academy of Sciences, 113, 9710–9715, https://doi.org/10.1073/pnas.1524766113, http://www.pnas.org/content/113/35/9710, 2016.

Zhang, X., Lohmann, G., Knorr, G., and Xu, X.: Different ocean states and transient characteristics in Last Glacial Maximum simulations and implications for deglaciation, Climate of the Past, 9, 2319–2333, 2013.

Zhang, X., Lohmann, G., Knorr, G., and Purcell, C.: Abrupt glacial climate shifts controlled by ice sheet changes, Nature, 512, 290–294, https://doi.org/10.1038/nature13592, 2014.

Zhu, J., Liu, Z., Brady, E., Otto-Bliesner, B., Zhang, J., Noone, D., Tomas, R., Nusbaumer, J., Wong, T., Jahn, A., and Tabor, C.: Reduced ENSO variability at the LGM revealed by an isotope-enabled Earth system model, Geophysical Research Letters, 44, 6984–6992, https://doi.org/10.1002/2017GL073406, 2017.

| Scenario | Northern Hemisphere | Antarctic coast | Globally uniform |
|---|---|---|---|
| fSL | based on sea-level changes (blue in Fig. 4f) | 0.0135 Sv - 140-130 ka | - |
| fIRD | based on IRD (red in Fig. 4f) | 0.0135 Sv - 140-130 ka | - |
| fIC | based on ice-sheet changes (black in Fig. 4f) | 0.0135 Sv - 140-130 ka | - |
| fSL2 | based on sea-level changes (blue in Fig. 4f) | triangular input with | - |
| | | max 0.15 Sv - 131-128 ka | - |
| fUN | - | - | based on sea-level changes |

**Table 2.** Freshwater scenarios for transient simulations of the PDG. Freshwater can be added in the NH and close to the Antarctic coast based on meltwater routing, or as a globally uniform flux. The Antarctic freshwater forcing in fSL2 is based on Goelzer et al. (2016a), but scaled down to obtain a total Antarctic contribution of 20 m sle.

| Forcing | Atmosphere | Ocean | Vegetation | Ocean biogeochemistry |
|---|---|---|---|---|
| Mole fraction of $CO_2^{(1)}$ | Surface air temperature$^{(X,Y)}$* | Sea Surface temperature$^{(X,Y)}$* | Biomes percentage$^{(X,Y)}$* | Dissolved Inorganic Carbon concentration |
| Mole fraction of $CH_4^{(1)}$ | Air temperature | Sea-water potential temperature | Primary production on land$^{(X,Y)}$ | Total Alkalinity |
| Mole fraction of $N_2O^{(1)}$ | Eastward wind* | Eastward windstress$^{(X,Y)}$ | Carbon mass in vegetation$^{(X,Y)}$ | Dissolved phosphate concentration |
| Ice-sheet mask$^{(X,Y)}$ | Northward wind* | Northward windstress$^{(X,Y)}$ | Carbon mass in litter pool$^{(X,Y)}$ | Dissolved nitrate concentration |
| Ice-sheet height$^{(X,Y)}$ | Geopotential height* | Eastward seawater velocity | Carbon mass in soil pool$^{(X,Y)}$ | Dissolved oxygen concentration |
| Land fraction$^{(X,Y)}$ | Total precipitation$^{(X,Y)}$* | Northward seawater velocity | Total carbon mass flux from$^{(X,Y)}$ | Dissolved inorganic silicon concentration |
| Sea-floor depth below geoid$^{(X,Y)}$ | Evaporation$^{(X,Y)}$* | Upward ocean mass transport$^{(X,Y)}$ | vegetation to litter | Dissolved iron concentration |
| Geographical location of$^{(X,Y)}$ | Total snow fall$^{(X,Y)}$* | Ocean meridional overturning | Total Carbon Mass Flux$^{(X,Y)}$ | Phytoplankton carbon concentration |
| meltwater input | Snow depth over land* | mass streamfunction | from litter to soil | Zooplankton carbon concentration |
| TOA incident shortwave$^{(X,Y)}$ | Surface Albedo$^{(X,Y)}$* | Sea-ice thickness$^{(X,Y)}$* | Total Carbon Mass Flux$^{(X,Y)}$ | Detrital Organic Carbon concentration |
| radiation | Meridional overturning | Sea-ice percentage$^{(X,Y)}$* | from vegetation to soil | Primary carbon production by phytoplankton* |
| TOA outgoing shortwave$^{(X,Y)}$ | mass streamfunction | Mixed layer depth$^{(X,Y)}$* | | Carbonate ion concentration |
| radiation | Relative humidity | Ocean meridional heat transport$^{(Y,B)}$ | | pH |
| | Surface sensible heat flux$^{(X,Y)}$ | Ocean heat content$^{(X,Y)}$ | | Delta $CO_2$ partial pressure$^{(X,Y)}$ |
| | Surface latent heat flux$^{(X,Y)}$ | Sea-surface salinity$^{(X,Y)}$* | | Surface downward flux of total $CO_2^{(X,Y)}$* |
| | Total cloud cover$^{(X,Y)}$ | Sea-water salinity | | |
| | Surface runoff over land $^{(X,Y)}$ | Sea-water potential density* | | |
| | Surface runoff over ocean$^{(X,Y)}$ | | | |

**Table 3.** Requested 1-dimensional$^{(1)}$, 2-dimensional (longitude, latitude$^{(X,Y)}$, latitude, basin$^{(Y,B)}$, or latitude,depth $^{(Y,Z)}$) and 3-dimensional (latitude, longitude, depth, or basin, longitude, depth, unmarked in the table) variables to be uploaded to the ESGF database. Most variables are requested as annual means, but * indicates that monthly outputs are requested. The CMIP6 conventions should be used for the variable names and units. See https://earthsystemcog.org/projects/wip/CMIP6DataRequest for details.

| Tracer interpretation | Core | Coordinates and depth (m) | $\varphi1$ (ka) | $\varphi2$ (ka) | $\varphi3$ (ka) | $\varphi4$ (ka) | $\varphi5$ (ka) | References |
|---|---|---|---|---|---|---|---|---|
| **Sea-level** | | | | | | | | |
| Sea-level | **Red Sea cores** | - | 137.0±0.7 *increases* | | 133.4±0.7 *main increase* | | 130.2±1 | Grant et al. (2012) This study |
| **Benthic $\delta^{13}$C** | | | | | | | | |
| North Atlantic intermediate-depth | **ODP983** | 60.40°N, 23.64°W 1984 m | 136.1±1.2 *weaker ventil.* | | | | 128.1±0.9 | Raymo et al. (2004) Barker et al. (2015) |
| ventilation | **ODP980** | 55.80°N, 14.11°W 2180 m | 137±1.9 | | | 128.6±1.8 | 127.6±1.3 | Oppo et al. (2006) |
| North Atlantic deep-water | **MD95-2042** | 37.80°N, 10.17°E 3146 m | | | | 131.0±1.4 *stronger ventil.* (T) | | Shackleton et al. (2003) |
| ventilation | **Stack of U1308 CH69-K09 and ODP 1063** | 49.88°N, 24.24°W, 3883 m 41.76°N, 47.35°W, 4100 m 33.69°N, 57.62°W, 4584 m | 135.9±2.0 | | | 130.3±1.6 | 129.2±1.4 | Hodell et al. (2008) Labeyrie et al. (1999) Deaney et al. (2017) |
| Southern Ocean deep-water ventilation | **MD02-2488** | 46.48°S, 88.02°E 3420 m | | | 131.9±2.1 *stronger ventil.* (U) | 130.2±2.2 *weaker ventil.* (V) | | Govin et al. (2009) Govin et al. (2012) |
| **Planktic $\delta^{18}$O and $\delta^{18}$Osw** | | | | | | | | |
| North Atlantic surface $\delta^{18}$O | **ODP980** | 55.80°N, 14.11°W 2180 m | | | | 130.0±1.3 | | Oppo et al. (2006) |
| *salinity* | **SU90-03** | 40.51°N, 32.05°W | | | | 131.0±1.1 | | CS98 |
| | **MD95-2042 ODP 976** | 37.80°N, 10.17°E 36.20°N, 4.31°E | | | 133.9±0.9 | 131.6±1.5 131.9±0.9 | | Shackleton et al. (2003) Martrat et al. (2014) |
| **Speleothem $\delta^{18}$Oc** | | | | | | | | |
| North Atlantic surface $\delta^{18}$O | **Corchia Cave, Italy** | 43.97°N, 13.0°E, 840 m a.s.l | | | 133.9±1.2 *NA meltwater input* (I) | 131.0±0.7 *NA meltwater paused* (J) | | Drysdale et al. (2009) Tzedakis et al. (2018) Marino et al. (2015) |
| **Mean ages for the beginning of $\varphi$1-$\varphi$5 from Tables 4 and 5** | | | | | | | | |
| Mean ages | | | 136.4±1.7 | 133.9±0.8 | 133.3±1.1 | 130.4±1.3 | 128.5±1.3 | |

**Table 4.**

**Table 4:** Key paleoproxy records of changes in ocean oxygen ($\delta^{18}$O) and carbon ($\delta^{13}$C) isotopic composition. The chronology of all paleoproxy records presented here is based on an alignment onto Corchia U-Th-based chronology (Table S1). Letters in brackets indicate the major changes identified in the paleoclimatic records shown in Figure 9. The dates of the major changes obtained through RAMPFIT or BREAKFIT are indicated with their associated uncertainties. Age uncertainties ($1\sigma$) reported include (1) the "internal" error of the event given by RAMPFIT (for most major changes) (Mudelsee, 2000), or BREAKFIT (major change dates highlighted with **) (Mudelsee, 2009), (2) the relative error related to the climatic alignment method for marine sediment records and (3) the absolute dating error of the reference time scale. Based on North Atlantic records, changes have been split into five phases ($\varphi$), with the date representing the beginning of each phase: $\varphi$1 is associated with the early phase of HS11 and MWP-2A, $\varphi$2 represents a pause within HS11, $\varphi$3 the main phase of HS11 and MWP-2B, $\varphi$4 the inception out of HS11 and $\varphi$5 full interglacial conditions. Implicitely the end of each phase corresponds to the beginning of the next one. For the sea-level record, the new chronology discussed in section 5 is used. CS98 refers to Chapman and Shackleton (1998). *ventil.* stands for *ventilation*.

| Tracer interpretation | Core, coordinates and depth/elevation | Chronology | φ1 (ka) | φ2 (ka) | φ3 (ka) | φ4 (ka) | φ5 (ka) | References |
|---|---|---|---|---|---|---|---|---|
| **Atm. CO₂ concentration** | | | | | | | | |
| Atm. $CO_2$ concentration | **EDC** 75.05°S, 123.19°E, 3233 m asl | Ice core AICC2012 chrono. | 137.8±2.7 *increases* | | | | 128.0±1.8 | Bereiter et al. (2015) |
| **SST** | | | | | | | | |
| North Atlantic *Summer* SST (FFA) | **ODP980** 55.80°N, 14.11°W, 2180 m depth | Alignment onto Corchia U-Th-based chrono. | | | | 129.7±1.3 | 128.7±1.3 | Oppo et al. (2006) |
| North Altantic *Summer* SST (FFA) | **SU90-03** 40.51°N, 32.05°W, 2475 m depth | Alignment onto Corchia U-Th-based chrono. | 136.9±1.6 *colder* [E] | | | 131±1 *warmer* [F] | | CS98 Cortijo et al. (1999) |
| W. Mediterr. Sea SST (Uk'37) | **ODP 976** 36.20°N, 4.31°E 1108 m depth | Alignment onto Corchia U-Th-based chrono. | 135.9±1.1 *colder* [A] | 134 ±1.1 [B] | 133.3** ±0.9 [C] | 131.4±0.9 *warmer* [D] | | Martrat et al. (2014) |
| Southern Ocean *Summer* SST (FFA) | **MD02-2488** 46.48°S, 88.02°E, 3420 m depth | Alignment onto AICC2012 | | | | 130±2.1** *max. warm* [R] | | Govin et al. (2009) Govin et al. (2012) |
| Southern Ocean SST (Mg/Ca) | **MD97-2120** 45.54°S, 174.92°E, 1210 m depth | Alignment onto AICC2012 | | | | 128.1±2.5** *max. warm* [S] | | Pahnke et al. (2003) |
| **Air temperature (SAT)** | | | | | | | | |
| Antarctic *Annual* SAT (Ice δD) | **EDC ice core** 75.1°S, 123.35°E, 3233 m a.s.l. | Ice core AICC2012 chrono. | 135.6±2.5 *(warmer)* [W] | | | 129.4±1.8** *max. warm* [X] | | Jouzel et al. (2007) |
| *SAT* Southern Europe ([Mg]) | **Corchia Cave, Italy** 43.97°N, 13.0°E, 840 m a.s.l | Alignment onto Corchia U-Th-based chrono. | | | | 131.0±0.7 *warmer* [K] | 128.6±0.6 *warm plateau* [L] | (Drysdale et al., 2019) |
| **Precipitation** | | | | | | | | |
| Vegetation/precipitation in Southern Europe | **Lago di Monticchio, Italy** 40.93°N, 15.58°E, 656 m a.s.l. | Independent absolute varve chronology | | | | | 127.2±1.6 *max. warm/wet* [G] | Brauer et al. (2007) |
| (Temperate tree pollen) | **Ioannina terr. sequence, Greece** 39.65°N, 20.91°E, 470 m a.s.l. | Orbital tuning | | | | 132.7±2.3 *wetter* [H] | | Tzedakis et al. (2003) |
| Intensity of Asian monsoon (δ¹⁸Oc) | **Chinese Caves** 25.28°N to 32.5°N 108.08-119.16°E 680 - 1900 m a.s.l. | Absolute U-Th-based chronology | 135.6±0.5 *drier* [M] | 133.7 ±0.5 [N] | 133.3** ±0.5 [O] | 128.9±0.1 *wetter* [P] | | Cheng et al. (2016) |
| Intensity of East Asian monsoon (grain size) | **Chinese loess** 35.62-37.14°N 103.20 - 109.85°E | Alignment onto Hulu-Sanbao U-Th-based chrono. | | | | 130.2±0.4 ∘ *wetter* [Q] | | Yang and Ding (2014) |
| **Mean ages for the beginning of φ1-φ5 from Tables 4 and 5** | | | | | | | | |
| Mean ages | | | 136.4±1.7 | 133.9±0.8 | 133.3±1.1 | 130.4±1.3 | 128.5±1.3 | |

**Table 5.**

**Table 5:** Same as Table 4 for key paleoproxy records of climatic and environmental changes through the PDG selected for their relatively high resolution. An indication of their climatic or environmental use is indicated in italic and the proxy used is shown in parenthesis in column 1. Age uncertainties ($1\sigma$) reported include (1) the "internal" error of the event given by RAMPFIT (for most major changes) (Mudelsee, 2000), or BREAKFIT (major change dates highlighted with **) (Mudelsee, 2009), (2) the relative error related to the climatic alignment method for marine sediment and Ioannina records and (3) the absolute dating error of the reference time scale. ○ No dating error is provided in Yang and Ding (2014). As a result, the stated error here only encompasses the "internal" error, it thus represents only a minimal error for the timing of the stacked grain size increase. The mean age representing the beginning of each phase ($\varphi$) is shown in the last row and is calculated from all the available dates within each phase shown in Tables 4 and 5. FFA stands for foraminifera faunal assemblages. CS98 refers to Chapman and Shackleton (1998). Based on North Atlantic records, changes have been split into five phases ($\varphi$), with the date representing the beginning of each phase: $\varphi1$ is associated with the early phase of HS11 and MWP-2A, $\varphi2$ represents a pause within HS11, $\varphi3$ the main phase of HS11 and MWP-2B, $\varphi4$ the inception out of HS11 and $\varphi5$ full interglacial conditions. Implicitely the end of each phase corresponds to the beginning of the next one.

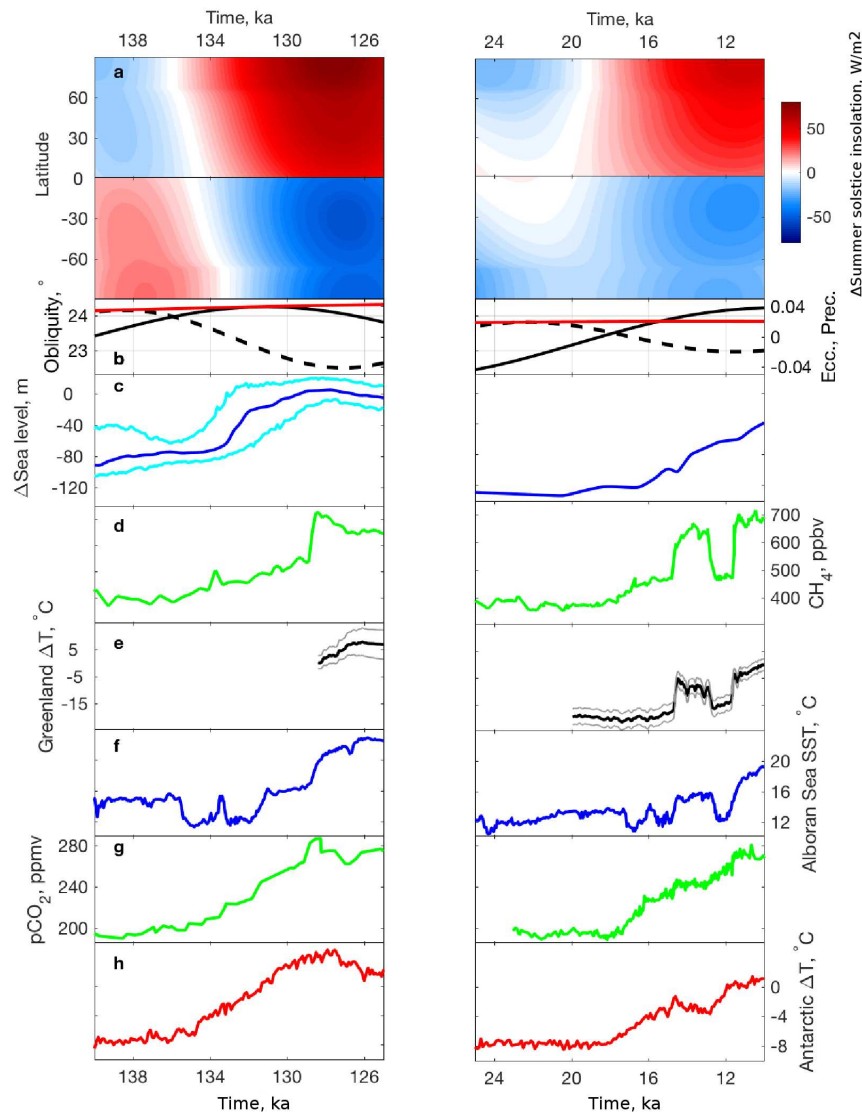

**Figure 1. Overview of (left) the penultimate and (right) the last deglaciations climatic and environmental evolutions: a)** Hovmöller diagram of summer solstice insolation anomalies (W/m$^2$). This corresponds to June 21st in the Northern Hemisphere and December 21st in the Southern Hemisphere; Timeseries of **b)** eccentricity (red), obliquity (solid black) and precession (dashed black) (Berger, 1978); **c)** Global mean sea level anomaly probability maximum (m, blue), (left) including its 95% confidence interval (cyan) (Grant et al., 2014), from Grant et al. (2012) on the original age model for the penultimate and (right) from Lambeck et al. (2014) for the last deglaciation. **d)** Atmospheric methane (CH$_4$) concentration as recorded in EDC ice core, Antarctica (Loulergue et al., 2008); **e)** Precipitation-weighted surface temperature reconstruction based on stable water isotopes from the Greenland NEEM ice core (left), and annual surface temperature composite reconstruction based on air nitrogen isotopes from the Greenland NEEM, NGRIP and GISP2 ice cores (Buizert et al., 2014); **f)** Alkenone-based (Uk'$_{37}$) SST reconstruction from ODP976 (Martrat et al., 2014); **g)** Atmospheric CO$_2$ concentration as recorded in Antarctic ice cores (left) EDC (Bereiter et al., 2015) and (right) WAIS divide (Marcott et al., 2014) on the WD2014 chronology (Buizert et al., 2015); **h)** Antarctic temperature anomalies relative to present day inferred from EDC ice core (Jouzel et al., 2007). Unless specified differently, all records are displayed on the AICC2012 timescale (Bazin et al., 2013; Veres et al., 2013) or a chronology coherent with AICC2012.

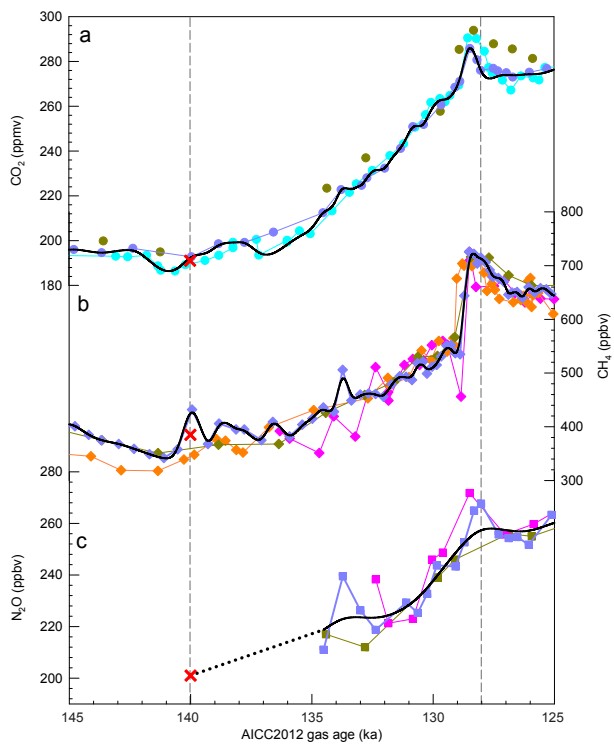

**Figure 2. Atmospheric greenhouse gas concentrations:** Atmospheric trace gases through the penultimate deglaciation from Antarctic ice cores displayed on the AICC2012 chronology (Bazin et al., 2013; Veres et al., 2013), and the spline that should be used to force the transient simulations (solid and dashed black lines). **a)** Atmospheric $CO_2$ concentrations from EDC (turquoise and blue) (Lourantou et al., 2010; Schneider et al., 2013) and TALDICE (green) (Schneider et al., 2013). **b)** Atmospheric $CH_4$ concentration from EDC (Loulergue et al., 2008) (blue), Vostok (Petit et al., 1999) (orange), TALDICE (Buiron et al., 2011) (green) and EDML (Capron et al., 2010) (pink). **c)** Atmospheric $N_2O$ concentration from EDC (Flückiger et al., 2002) (blue), EDML (Schilt et al., 2010) (pink) and TALDICE (Schilt et al., 2010) (green). Due to in-situ production within the ice-sheet, no accurate $N_2O$ measurements are available beyond 134.5 ka. Between 140 and 134.5 ka $N_2O$ should increase linearly from 201 ppb to 218.74 ppb (dashed black line). Red crosses indicate the 140 ka spin-up values for $CO_2$, $CH_4$ and $N_2O$ concentrations (i.e. 191 ppm, 385 ppb and 201 ppb, respectively).

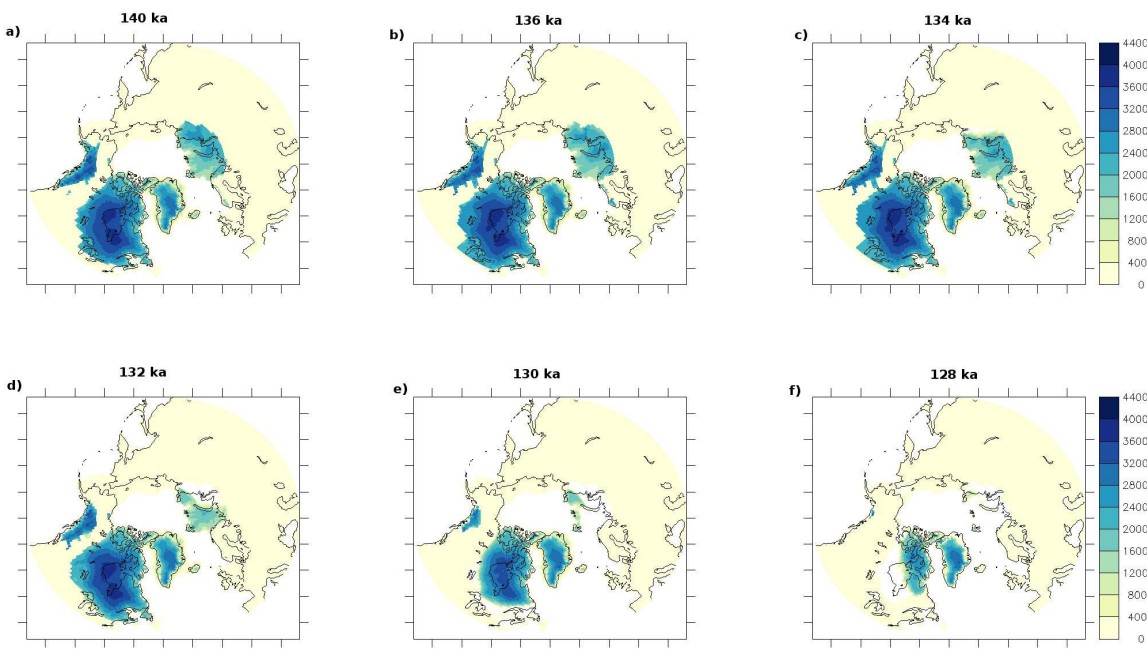

**Figure 3. NH ice-sheets** elevation (m) at **a)** 140, **b)** 136, **c)** 134, **d)** 132, **e)** 130 and **f)** 128 ka from the combined ice-sheet forcing: as simulated by the IcIES-MIROC model (Abe-Ouchi et al., 2013) for the North American and Eurasian ice-sheets, and as simulated by the Glacial Systems Model (GSM) (e.g. Tarasov et al., 2012) for the Greenland ice-sheet. Elevation is shown where the ice-mask is greater than 0.5.

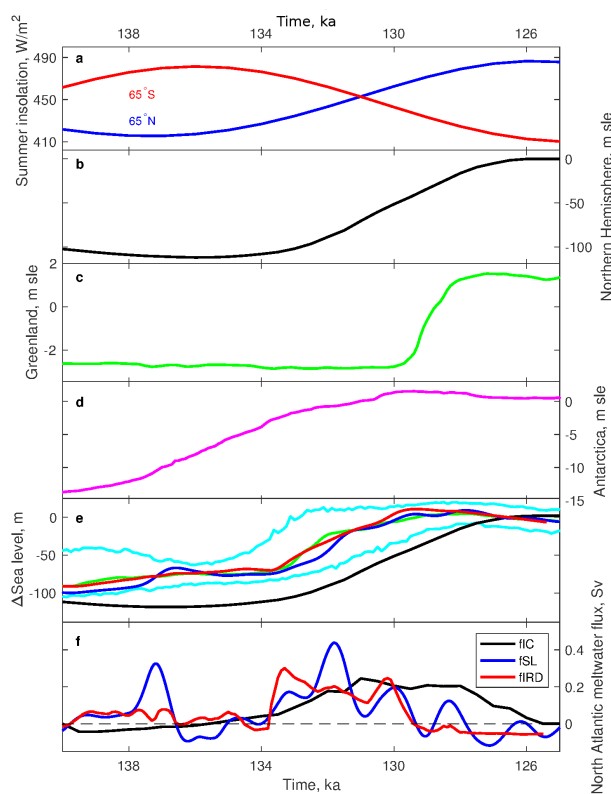

**Figure 4.** Timeseries of **a)** Summer solstice insolation at 65°N and 65°S, **b-d)** global sea level equivalent (m) of changes in **b)** NH (Abe-Ouchi et al., 2013), **c)** Greenland (Tarasov et al., 2012) and **d)** Antarctic ice-sheets (Briggs et al., 2014), **e)** global sea level (m) change estimated from Red Sea records (Grant et al., 2014) (green) with the 95% probability interval (cyan), sea-level record with an adjusted age scale as described in section 5 (blue), and as estimated from the continental ice-sheets simulations (black). The red line shows the changes in global sea-level that would be obtained by adding the meltwater flux, fIRD, described in **f** plus an Antarctic contribution of 13 m. **f)** Possible North Atlantic meltwater flux (Sv) scenarios: estimated from the disintegration of NH ice-sheets as shown in **b, c** (Tarasov et al., 2012; Abe-Ouchi et al., 2013) (black), estimated from the global sea-level change on the revised age-scale (this study, blue), and scaled from the North Atlantic and Norwegian Sea IRD records (red, shown in Fig. 8b).

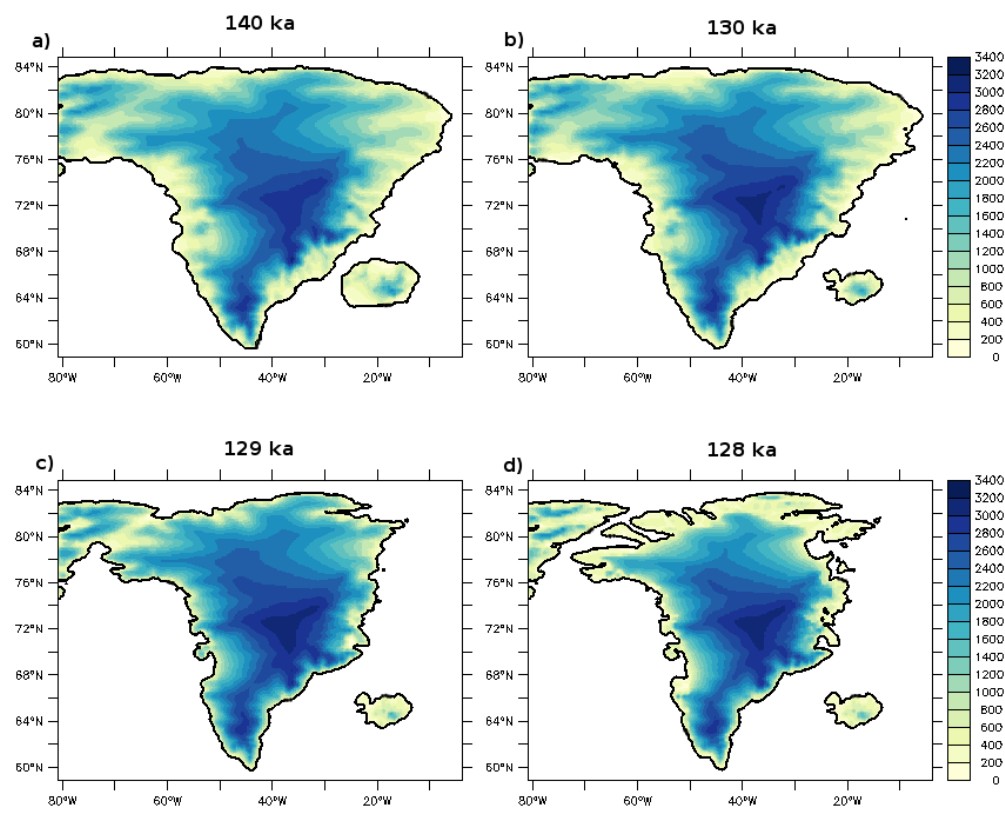

**Figure 5. Greenland ice-sheet** elevation (m) at **a)** 140 ka, **b)** 130 ka, **c)** 129 ka, and **d)** 128 ka as simulated by the GSM (e.g. Tarasov et al., 2012). The 0 elevation contour is in black.

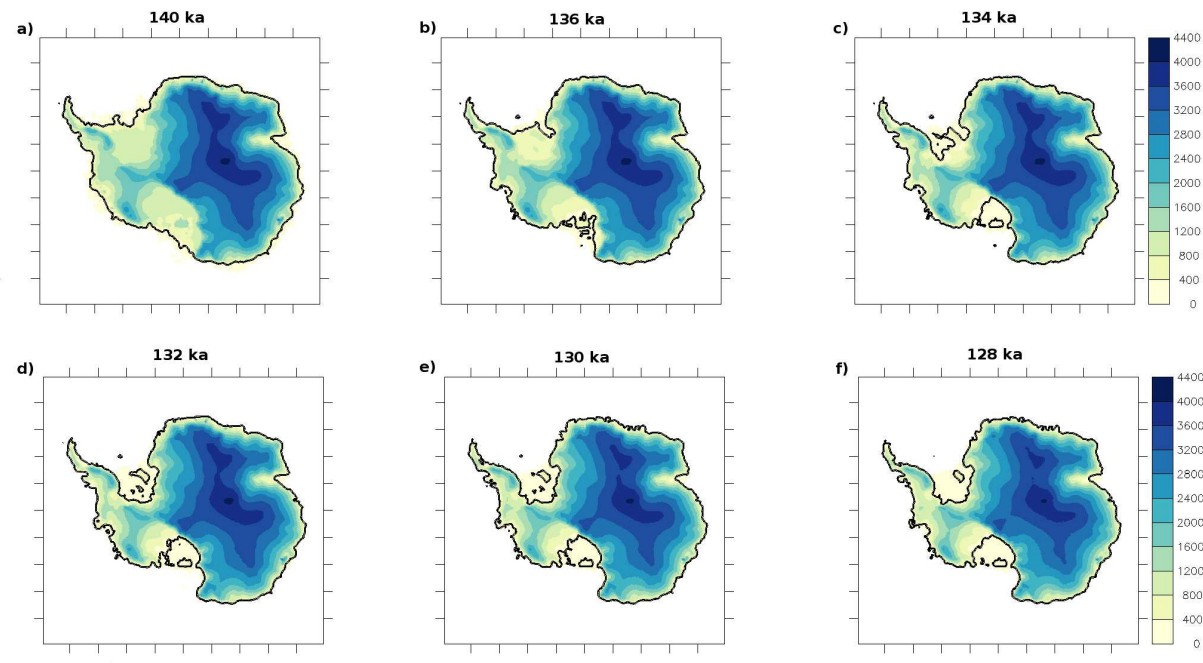

**Figure 6. Antarctic ice-sheet** elevation at **a)** 140 ka, **b)** 136 ka, **c)** 134 ka, **d)** 132 ka, **e)** 130 ka and **f)** 128 ka as simulated by the GSM (e.g. Briggs et al., 2014). Grounding lines are in black.

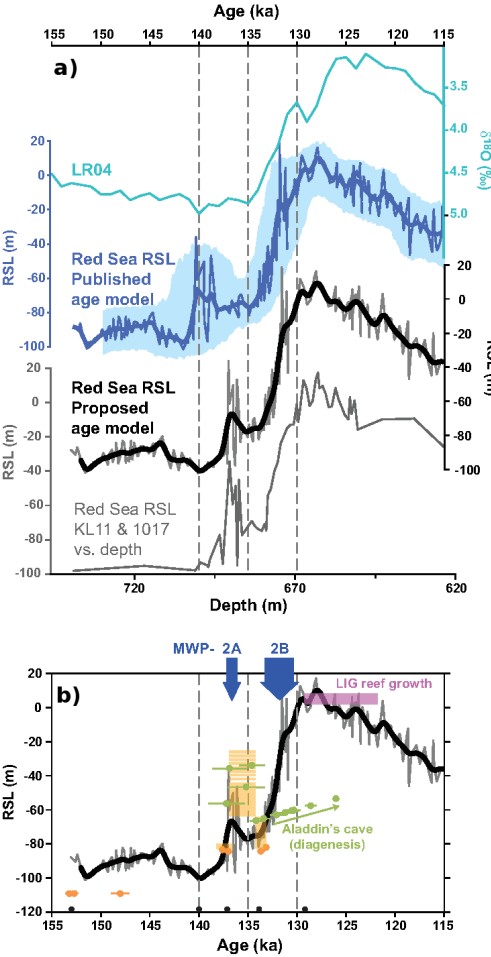

**Figure 7. Sea-level changes across the PDG: a)** Stacked LR04 benthic $\delta^{18}$O curve (turquoise) (Lisiecki and Raymo, 2005), Red Sea Relative sea level (RSL) data (1 ka moving Gaussian filter in dark blue line and 95% probability interval in light blue) on the age model from Grant et al. (2012) and on the new age model (this study, data in grey, 1 ka moving Gaussian filter in black); RSL curve inferred from Red sea Geo-KL11 and MD92-1017 cores on a depth scale (dark grey) (Grant et al., 2012). **b)** Tahiti (orange dots) (Thomas et al., 2009) and Huon Peninsula (green dots) (Esat et al., 1999) corals superimposed onto the Red Sea RSL curve displayed on the new age model (black). Orange bars indicate the range of the paleowater depth estimates from Tahiti corals. Coral data have been updated to new decay constants (Cheng et al., 2013) (Tables S3, S4). Black dots indicate the tie points defined so that the timing of MWP-2A in the Red Sea RSL is consistent with the absolutely-dated Tahiti and Huon Peninsula coral data (Table S5). The pink box represents the timing and range of RSL during the main phase of LIG reef growth (129-122 ka) (Dutton and Lambeck, 2012). Vertical dashed grey lines indicated the 140, 135 and 130 ka time intervals on both panels.

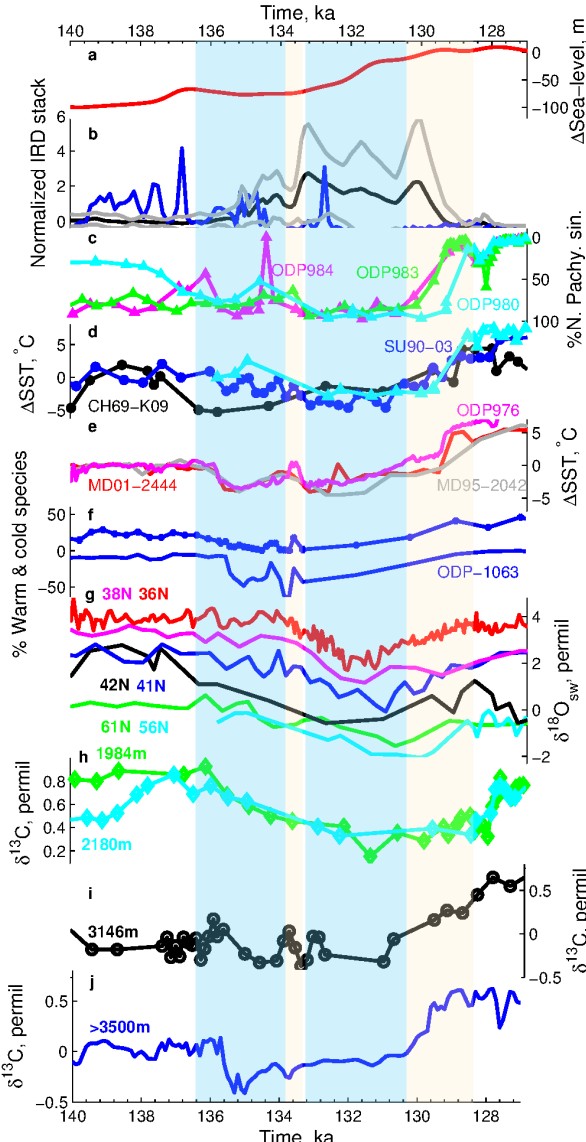

**Figure 8. Selection of RSL and North Atlantic marine sediment records: a)** Red Sea RSL record (Grant et al., 2012), with revised age-scale as described in Section 5. **b)** Normalized IRD stack (black) of cores CH69-K09 (Labeyrie et al., 1999), ODP980 (Oppo et al., 2006), ODP983 (Barker et al., 2015), ODP984 (Mokeddem et al., 2014), ODP1063 (Deaney et al., 2017) and SU90-03 (Chapman and Shackleton, 1998) with $\pm 1\sigma$ (grey envelope). Normalized IRD record of MD95-2010 (blue) (Risebrobakken et al., 2006). **c)** *N. Pachyderma sin.* (%) in cores ODP980 (cyan) (Oppo et al., 2006), ODP983 (green) (Barker et al., 2015) and ODP984 (magenta) (Mokeddem and McManus, 2016); **d)** Estimated summer SST anomalies (°C) in cores ODP980 (cyan) , CH69-K09 (black) (Labeyrie et al., 1999) and SU90-03 (blue) (Chapman and Shackleton, 1998); **e)** Uk'37 SST anomalies (°C) in cores MD01-2444 (red) (Martrat et al., 2007), ODP976 (magenta) and MD95-2042 (grey) (Martrat et al., 2014); **f)** % of warm (circles) and cold (axis reversed) species in core ODP1063 (Deaney et al., 2017); **g)** Estimated $\delta^{18}O_{sw}$ (‰) in cores ODP976 (red), MD95-2042 (magenta) (Martrat et al., 2014), SU90-03 (blue) (Chapman and Shackleton, 1998), CH69-K09 (black) (Labeyrie et al., 1999), ODP984 (green) (Mokeddem and McManus, 2016), ODP980 (cyan) (Oppo et al., 2006); Benthic $\delta^{13}C$ (‰) in cores **h)** ODP983 (green) (Lisiecki and Raymo, 2005) and ODP980 (cyan) (Oppo et al., 2006), **i)** MD95-2042 (black) (Martrat et al., 2014), **j)** a $\delta^{13}C$ stack of cores U1308, CH69-K09 and ODP1063 (Labeyrie et al., 1999; Hodell et al., 2008; Deaney et al., 2017). All the age models have been revised as described in the methods and in Govin et al. (2015).

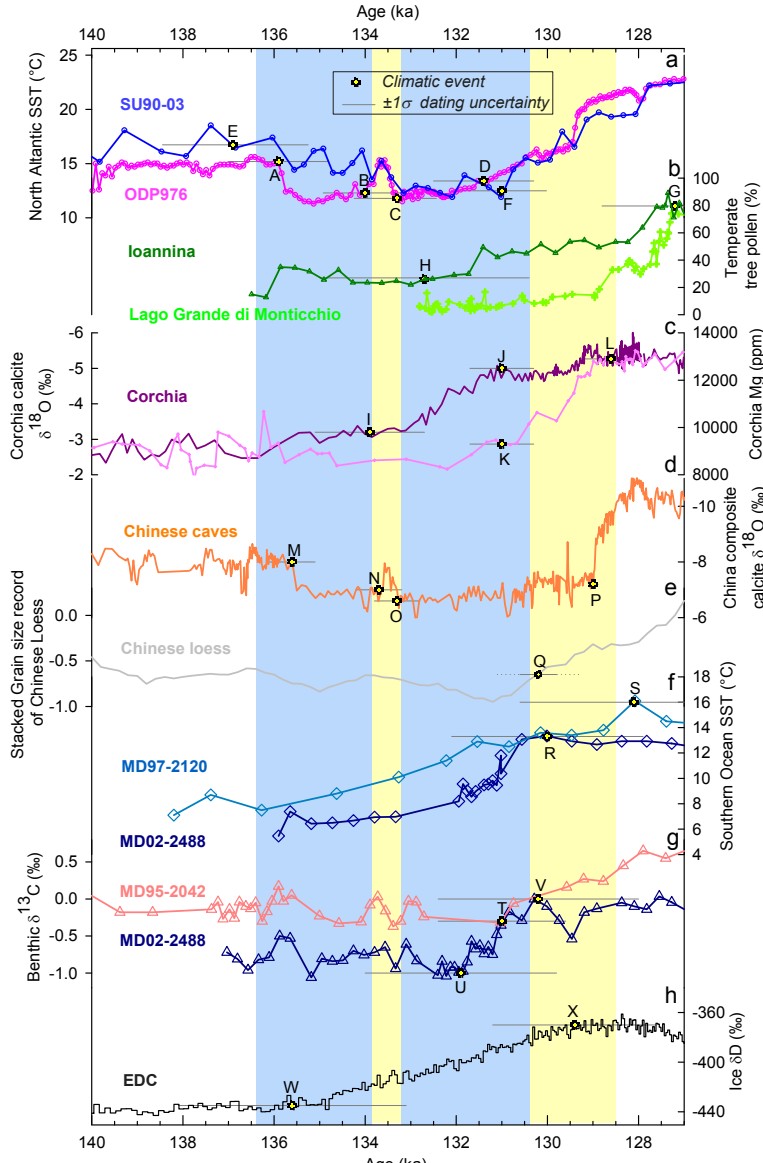

**Figure 9. Selection of paleorecords illustrating climatic and environmental changes between 140 and 127 ka. a)** SST records from North Atlantic cores ODP-976 (pink) (Martrat et al., 2014) and SU90-03 (blue) (Chapman et al., 2000). **b)** Percentage of temperature tree pollen from the Greek Ioannina sequence (dark green) (Tzedakis et al., 2003) and from the Italian Lago Grande di Monticchio (LGdM) sequence (light green) (Brauer et al., 2007). **c)** Composite speleothem calcite $\delta^{18}O$ record from the Italian Corchia Cave speleothem (purple) and Corchia CD Mg concentration (Drysdale et al., 2007, 2019) **d)** Composite calcite $\delta^{18}O$ record from the Chinese Hulu, Sanbao and Dongge Caves speleothems (orange) (Cheng et al., 2016). **e)** Staked grain size record of Chinese Loess (light grey) (Yang and Ding, 2014). **f)** SST records from Southern Ocean cores MD97-2120 (light blue) (Pahnke et al., 2003) and MD02-2488 (dark blue) (Govin et al., 2012). **g)** $\delta^{13}C$ record from North Atlantic core MD95-2042 (light pink) (Shackleton et al., 2003) and Southern Ocean core MD02-2488 (dark blue) (Govin et al., 2012). **h)** Antarctic ice $\delta D$ record from EPICA Dome C (Jouzel et al., 2007) on AICC2012 chronology (black) (Bazin et al., 2013; Veres et al., 2013). Letters in brackets and yellow crosses indicate the major changes identified in the paleoclimatic records, and their $\pm 1\sigma$ dating uncertainty (horizontal grey bars, Tables 4 and 5). Chronologies and associated references are detailed in Tables 4 and 5. No details on the dating errors attached to the Chinese Loess stacked grain size record are provided in (Yang and Ding, 2014), the $1\sigma$ error should thus be treated as an underestimated dating error.