# Peer review of "The penultimate deglaciation: protocol for PMIP4 transient numerical simulations between 140 and 127 ka, version 1.0"

_Geoscientific Model Development, 2019_

## Referee Comment (RC1) · Anonymous Referee #1 · 21 Apr 2019

As one of the reviewers of the original submission of this manuscript for 'Climate of the Past', I have already evaluated the paper once before transfer to GMD. My advice was publication with minor revisions. Reexamining the paper now, I confirm this judgment and note that the few suggestions I had, have all been addressed in the present version of the manuscript. I came across a few more minor points that can be addressed without the need for me to see the paper again.

Minor points:

P1 L2 Consider defining an abbreviation for "penultimate glacial maximum", which occurs several time throughout the manuscript (p8, p20, p25). Note that in one instance

it is referred to as"(MIS 6)".

P2 L21 The interglacial states are different, which is important motivation for your experiment. Consider reformulation: Each of these long glacial periods were followed by relatively rapid multi-millennial-scale warmings into consecutive interglacial state*s*

P2 L22 Consider explicitly adding 'deglaciations' here instead of line 24-25: These glacial-interglacial transitions, also called deglaciations, represent the largest natural global warmings and large-scale climate reorganisations across the Quaternary.

P2 L23 Avoid confusion with term "climate sensitivity": Hence, they provide a great opportunity to study the interaction between the different components of the Earth System and climate*'s* sensitivity to changes in radiative forcing.

P2 L24 "radiative forcing". At this point in the text, it has not yet been made clear that glacial-interglacial transitions are forced by orbital changes. So this comes a bit unexpected. consider reordering.

P2 L30 Consider moving (Fig. 1) reference to the end of the sentence to avoid triple brackets.

P2 L32 You are back to discussing five deglaciations, but we are looking at figure 1 now. Consider revising.

P3 L4 Add 'concurrent' after "supported by the".

P3 L11 "in the *Northern* North Atlantic".

P3 L19 "Antarctic sea ice cover *change*"

P3 L24 "the last five deglaciations" or "the last two deglaciations".

P4 L16 You may want to somehow distinguish between the left and the right side in figure 1 to make your references more specific.

P4 L21 Replace 'find no evidence' by 'provide no evidence'.

P4 L27 Suggest to change the order in this sentence to: The eustatic sea-level during MIS 6 is estimated at âĹij90-100 m lower than present-day (Rabineau et al., 2006; Grant et al., 2012; Rohling et al., 2017), with a relatively large uncertainty range (Rohling et al., 2017). This compares to >130 m lower than present-day during the LGM (Austermann et al., 2013; Lambeck et al., 2014).

P4 L30 I don't think you have already establish that the Holocene follows the last deglaciation?

P5 L19 This sentence suggests that climate, vegetation and carbon cycle respond to changes in the oceanic circulation, but they also respond directly to ice sheet disintegration. Suggest to remove "and thus".

P5 L22 add 'forcing' after 'external' and 'dynamics' after 'internal'. Remove 'forcings'.

P6 L6 Remove 'the' before 'the'.

P6 L8 Reformulate " in concert with paleoclimate records"

P6 L9 Add "with coupled AOGCMs" after 140 to 127.

P6 L9 Suggest to start a new sentence: "This experiment provides a link" ...

P6 l11 Clarify why the forcings at 127 are different. Maybe this should be explained earlier on.

P6 L11-13 Consider adding section numbers to this manuscript overview.

P9 L13 Remove "shallow" here as it is mentioned in l14.

P9 L16 Reference Payne 1999 is undefined. Maybe Payne, AJ., 1999, A thermomechanical model of ice flow in West Antarctica : Climate Dynamics. 15, p.115 - 125. However, if this is a Weertman type sliding law, a reference to Weetman seems more appropriate, especially because Payne 1999 is for an Antarctic case. Please also check all the other references once more.

P10 L18 Still unclear what the first 240 ka model run looks like and why that is needed. Is it identical to the first run?

P10 L21 Add "a" before '2-way'.

P10 L30 "Greenland loses its glacial grounded ice volume" is still confusing to me. It is not really possible to spatially distinguish glacial ice from the rest. I think you try to say that in this period the GrIS loses the ice mass in excess of the total present-day value and more. Maybe "In this simulation, the main phase of Greenland deglaciation occurs between 130 and 127 ka, during which Greenland first loses an ice mass of âĹij2.9 m sle in excess of the total pre-industrial value, and then an additional 1.5 m sle."

P27 Data availability

I would reserve this section to information on where the forcing data can be accessed and move the instruction for the participating groups to elsewhere (e.g. section 8.4). I am not sure if a wiki page is acceptable as data source for GMD under the new editorial, but leave it to the editor to negotiate that.

P45 Table3

- Is it clear which variables are XY, XZ and so on, or should this be clarified in the table?

P50 Fig 1

- Is it necessary to clarify that the summer solstice happens at different moments in SH and NH?

- Could you clarify in the text what motivates the relative alignment of these time windows?

- "Unless specified *differently*, all ...".

P54 Fig 5

- Correct some plotting artefacts:

- Close the grounding lines, e.g. in the NE in b)

- What does an inland closed circle GL mean? Does the calving flux go into a lake?

P55 Fig 6

- Can the plotting artefact at the date line be avoided?

- remove '0' label on the grounding line contour.

---

## Referee Comment (RC2) · Julia Hargreaves (Referee) · 28 May 2019

This is a very short review to state that, in my opinion, the authors have responded sufficiently to all the points raised by Anonymous Reviewer 2 in the open review at Climate of the Past (https://www.clim-past-discuss.net/cp-2018-106/).

The original review at CP can be found here: https://www.clim-past-discuss.net/cp-2018-106/cp-2018-106-RC2.pdf And the authors' response here: https://www.clim-past-discuss.net/cp-2018-106/cp-2018-106-AC4-supplement.pdf

---

## Editor Comment (EC1) · Julia Hargreaves (Editor) · 28 May 2019

The original version of this manuscript was reviewed at Climate of the Past (https://www.clim-past-discuss.net/cp-2018-106/ doi:10.5194/cp-2018-106). As the online review there shows, two positive reviews and some interesting comments from the community were received and responded to. The coordinator of the PMIP4 special issue in GMD and CP decided that the manuscript should have been submitted to GMD rather than CP. This is because it is very clearly a model experiment description. Therefore, the paper was withdrawn from CP, revised in accordance with the initial review at CP, and then submitted to GMD. The authors helpfully provided me with the diff file for

their revision compared to the version reviewed at CP. After some further revision at the initial editor's decision stage to make sure that the manuscript more or less complied with GMD requirements, I decided that it was sufficient to recall the two original CP reviewers. Both agreed to review, but only one has actually provided a review. I have decided to draw a line under the other reviewer's repeated promises to respond soon, and have now checked carefully through their review at CP (reviewer 2 in the CP version), and the author's response. I conclude (as you see in my short review!) that the authors adequately responded to that reviewer.

Therefore, please revise the manuscript in accordance with the new comments from Reviewer 1 and my additional comments below, which were mostly brought to my attention by Reviewer 1's original review at CP.

1. In order for "recommend" to be interpreted as "should" you need to include a sentence in the manuscript where you state that this is your new definition of this word. Otherwise it will be misinterpreted. So, either define "recommend", or change "recommend" to "should" where relevant.

2. Some information on the merger of the ice sheets was provided in the response to the reviewer, but I do not see this in the manuscript. Readers need to know what you have done. As well as a clear description, can you also provide the individual ice sheets and the code as well as the final result?

3. A similar theme for the Red Sea records... I am not convinced that the sea-level evolution result is reproducible from what you have written, so providing the associated data and code used to derive the new result would be very useful.

3. The data availability section needs updating to include the information that the forcings are included in the supplement. All forcings should be in the manuscript itself, or in the supplement, or on a public repository with a unique and persistent identifier (such as a DOI).

---

## Author Comment (AC1) · 12 Jun 2019

As one of the reviewers of the original submission of this manuscript for 'Climate of the Past', I have already evaluated the paper once before transfer to GMD. My advice was publication with minor revisions. Reexamining the paper now, I confirm this judgment and note that the few suggestions I had, have all been addressed in the present version of the manuscript. I came across a few more minor points that can be addressed without the need for me to see the paper again.

We thank the Reviewer for their positive comment and careful review, which helped improve the manuscript. Please find our answer to comments in blue as well as suggested text changes in green.

Minor points:

P1 L2 Consider defining an abbreviation for "penultimate glacial maximum", which occurs several time throughout the manuscript (p8, p20, p25). Note that in one instance it is referred to as"(MIS 6)".

We are following the reviewer's suggestion and using the abbreviation PGM for "penultimate glacial maximum" throughout the manuscript. MIS 6 was also replaced by PGM.

P2 L21 The interglacial states are different, which is important motivation for your experiment. Consider reformulation: Each of these long glacial periods were followed by relatively rapid multi-millennial-scale warmings into consecutive interglacial state*s*

This has been amended, thank you.

P2 L22 Consider explicitly adding 'deglaciations' here instead of line 24-25: These glacial-interglacial transitions, also called deglaciations, represent the largest natural global warmings and large-scale climate reorganisations across the Quaternary.

The beginning of the sentence L.22 was amended as follows:

"These deglaciations represent the largest…"

And the sentence L. 24-25 was taken out.

P2 L23 Avoid confusion with term "climate sensitivity": Hence, they provide a great opportunity to study the interaction between the different components of the Earth System and climate*'s* sensitivity to changes in radiative forcing.

This has been amended, thank you.

P2 L24 "radiative forcing". At this point in the text, it has not yet been made clear that glacial-interglacial transitions are forced by orbital changes. So this comes a bit unexpected. consider reordering.

The first paragraph was re-ordered so that the importance of orbital changes is stated earlier on p2, L. 23.

P2 L30 Consider moving (Fig. 1) reference to the end of the sentence to avoid triple brackets.

This has been amended, thank you.

P2 L32 You are back to discussing five deglaciations, but we are looking at figure 1 now. Consider revising.

It is true that Figure 1 only focuses on the last two deglaciations, which are then discussed in details in the text. However, the two first paragraphs present a broad overview of the deglaciations of the last 450 ka. Therefore, we have kept the mention to the last five deglaciations.

P3 L4 Add 'concurrent' after "supported by the".

"concurrent" has been added.

P3 L11 "in the *Northern* North Atlantic".

"northern" has been added.

P3 L19 "Antarctic sea ice cover *change*"

We taken out at "changes" in front of "ocean circulation", as the "changes in" L. 18 refers to all the processes discussed, i.e. solubility, alkalinity, iron….

P3 L24 "the last five deglaciations" or "the last two deglaciations".

We have modified the sentence so that it reads: "the last two deglaciations".

P4 L16 You may want to somehow distinguish between the left and the right side in figure 1 to make your references more specific.

The reference to the figure now read "Fig. 1, right".

P4 L21 Replace 'find no evidence' by 'provide no evidence'.

This has been amended, thank you.

P4 L27 Suggest to change the order in this sentence to: The eustatic sea-level during MIS 6 is estimated at 90-100 m lower than present-day (Rabineau et al., 2006; Grant et al., 2012; Rohling et al., 2017), with a relatively large uncertainty range (Rohling et al., 2017). This compares to >130 m lower than present-day during the LGM (Austermann et al., 2013; Lambeck et al., 2014).

The sentence has been amended as suggested, thank you.

P4 L30 I don't think you have already establish that the Holocene follows the last deglaciation?

The sentence was amended as follows: "The LIG also bears significant differences to the interstadial that followed the last deglaciation, i.e. the Holocene."

P5 L19 This sentence suggests that climate, vegetation and carbon cycle respond to changes in the oceanic circulation, but they also respond directly to ice sheet disintegration. Suggest to remove

"and thus".
This has been amended.

P5 L22 add 'forcing' after 'external' and 'dynamics' after 'internal'. Remove 'forcings'.
This has been amended.

P6 L6 Remove 'the' before 'the'.
This has been amended.

P6 L8 Reformulate " in concert with paleoclimate records"
We have taken out " in concert with paleoclimate records".

P6 L9 Add "with coupled AOGCMs" after 140 to 127.
This has been amended.

P6 L9 Suggest to start a new sentence: "This experiment provides a link" ...
This has been amended.

P6 l11 Clarify why the forcings at 127 are different. Maybe this should be explained earlier on.
The sentence: "albeit with some differences in the ice-sheet and meltwater forcings at 127 ka." was deleted. The difference between the end of the penultimate deglaciation experiments and the lig127k experiment is mentioned in section 7 (p19, L.29-31):
"Finally, the proposed experiment will provide a link to the PMIP4 transient simulation of the LIG (127 to 121 ka) as well as the PMIP4 127 ka timeslice experiment (liig127k) (Otto-Bliesner et al., 2017), even though the protocol of the LIG experiments includes pre-industrial continental ice-sheets."

P6 L11-13 Consider adding section numbers to this manuscript overview.
Section numbers have been added to this overview.

P9 L13 Remove "shallow" here as it is mentioned in l14.
This has been amended.

P9 L16 Reference Payne 1999 is undefined. Maybe Payne, AJ., 1999, A thermomechanical model of ice flow in West Antarctica : Climate Dynamics. 15, p.115 - 125. However, if this is a Weertman type sliding law, a reference to Weetman seems more appropriate, especially because Payne 1999 is for an Antarctic case. Please also check all the other references once more.
The reference to Payne 1999, Climate Dynamics has been added.
In the IcIES model the sliding velocity is related to the gravitational driving stress, as defined in Payne, 1999.

P10 L18 Still unclear what the first 240 ka model run looks like and why that is needed. Is it identical to the first run?

In order to get a good representation of the evolution of the Greenland ice-sheet over the penultimate deglaciation, a proper 140 ka state is needed. Given that with active bed thermodynamics (down to 4km), the thermodynamic equilibration timescale is greater than 100 kyr for GRIS, the most appropriate method is to start the run at 240ka.

The text has been amended as follows:
"Given that with active bed thermodynamics (down to 4km), the thermodynamic equilibration timescale is greater than 100 ka for the Greenland ice-sheet, the most appropriate method is to start the run during the previous interglacial period. Therefore, the model runs start at 240 ka with present-day ice and bedrock geometry..."

P10 L21 Add "a" before '2-way'.
This has been amended.

P10 L30 "Greenland loses its glacial grounded ice volume" is still confusing to me. It is not really possible to spatially distinguish glacial ice from the rest. I think you try to say that in this period the GrIS loses the ice mass in excess of the total present-day value and more. Maybe "In this simulation, the main phase of Greenland deglaciation occurs between 130 and 127 ka, during which Greenland first loses an ice mass of 2.9 m sle in excess of the total pre-industrial value, and then an additional 1.5 m sle."
The sentence has been amended as suggested.

P27 Data availability I would reserve this section to information on where the forcing data can be accessed and move the instruction for the participating groups to elsewhere (e.g. section 8.4). I am not sure if a wiki page is acceptable as data source for GMD under the new editorial, but leave it to the editor to negotiate that.
We have removed the sentence with respect to uploading results to the ESGF database, as it is already included in section 6.3.
All the forcings are either included in the supplement (freshwater, ice-sheet forcing), already include a DOI (GHGs), or have now been assigned a DOI (ice-sheet). The data availability section now reads:
"The combined ice-sheet and meltwater scenarios are available in the Supplement. The GHG data can be found at \url{https://doi.org/10.1594/PANGAEA.871273}. The combined ice-sheet, as well as the Northern Hemispheric and Greenland ice-sheets are also publicly available on the Research Data Australia repository (doi:).
In addition, all the forcing files as well as the paleo-data described in the manuscript are available on the PMIP4 wiki: \url{https://pmip4.lsce.ipsl.fr/doku.php/exp_design:degla_t2}."

P45 Table3 - Is it clear which variables are XY, XZ and so on, or should this be clarified in the table?

A system of superscript was inserted in the table with (1) for 1-dimensional variables, and (X,Y) and (Y,B) for (longitude, latitude) and (latitude, basin) variables, respectively.

P50 Fig 1

- Is it necessary to clarify that the summer solstice happens at different moments in SH and NH?

A sentence has been added in the legend:

"This corresponds to June 21st in the Northern Hemisphere and December 21st in the Southern Hemisphere;"

- Could you clarify in the text what motivates the relative alignment of these time windows?

The sentence below has been added in the Introduction (p3, L. 23-24):

"Figure 1 shows the evolution of key variables across 15 ka, from glacial maxima to peak interglacial conditions."

- "Unless specified *differently*, all ...".

This has been amended.

P54 Fig 5 - Correct some plotting artefacts:

- Close the grounding lines, e.g. in the NE in b)

- What does an inland closed circle GL mean? Does the calving flux go into a lake?

We have now corrected the plotting artifacts in the Greenland figure.

P55 Fig 6 - Can the plotting artefact at the date line be avoided? - remove '0' label on the grounding line contour.

The artefact at the date line has been fixed and the 0 'label' removed.

---

## Author Comment (AC2) · 12 Jun 2019

The original version of this manuscript was reviewed at Climate of the Past (https://www.clim-past-discuss.net/cp-2018-106/ doi:10.5194/cp-2018-106). As the online review there shows, two positive reviews and some interesting comments from the community were received and responded to. The coordinator of the PMIP4 special issue in GMD and CP decided that the manuscript should have been submitted to GMD rather than CP. This is because it is very clearly a model experiment description. Therefore, the paper was withdrawn from CP, revised in accordance with the initial review at CP, and then submitted to GMD. The authors helpfully provided me with the diff file for their revision compared to the version reviewed at CP. After some further revision at the initial editor's decision stage to make sure that the manuscript more or less complied with GMD requirements, I decided that it was sufficient to recall the two original CP reviewers. Both agreed to review, but only one has actually provided a review. I have decided to draw a line under the other reviewer's repeated promises to respond soon, and have now checked carefully through their review at CP (reviewer 2 in the CP version), and the author's response. I conclude (as you see in my short review!) that the authors adequately responded to that reviewer. Therefore, please revise the manuscript in accordance with the new comments from Reviewer 1 and my additional comments below, which were mostly brought to my attention by Reviewer 1's original review at CP.

We thank the Editor for facilitating the transfer of the manuscript from Climate of the Past to Geoscientific Model Development. Please find our answer to comments in blue as well as suggested text changes in green.

1. In order for "recommend" to be interpreted as "should" you need to include a sentence in the manuscript where you state that this is your new definition of this word. Otherwise it will be misinterpreted. So, either define "recommend", or change "recommend" to "should" where relevant.

We have changed "recommend" into should.

2. Some information on the merger of the ice sheets was provided in the response to the reviewer, but I do not see this in the manuscript. Readers need to know what you have done. As well as a clear description, can you also provide the individual ice sheets and the code as well as the final result?

Some information on the merger has been added in section 4.1:

"The merger involves no extra smoothing, beyond that inherent in the GIA solver which involves transformation to spherical harmonics. The merger involves a simple masking operation with the mask boundary through Nares Strait, Baffin Bay, Davis Strait, and the Labrador Sea. Examination of the resultant topography shows small merger artifacts around Nares Strait ranging to a few hundred metres in elevation difference."

The combined ice-sheet, as well as the individual Northern Hemispheric ice-sheet and the Greenland ice-sheet have been uploaded onto Research Data Australia repository

at . http://handle.unsw.edu.au/1959.4/resource/collection/resdatac_874/1

The data will be published and the doi assigned on June 21st.

3. A similar theme for the Red Sea records... I am not convinced that the sea-level evolution result is reproducible from what you have written, so providing the associated data and code used to derive the new result would be very useful.

The text, section 6 details the methodology taken to generate the new sea-level chronology. All the information necessary to redo the new chronology is in the Supplementary table 5, which shows the new tie points as well as associated comments.

4. The data availability section needs updating to include the information that the forcings are included in the supplement. All forcings should be in the manuscript itself, or in the supplement, or on a public repository with a unique and persistent identifier (such as a DOI)

The supplementary information includes the combined ice-sheet forcing, the meltwater scenarios, as well as the new sea-level chronology. The greenhouse gas forcing is already published and has a DOI that is provided. In addition, the combined ice-sheet forcing as well as the individual ice-sheets have been uploaded to Research Data Australia repository, where they will be publicly available and a DOI will be given on June 21st.

The data availability section has been updated to point the reader to the forcing files:

"The combined ice-sheet and meltwater scenarios are available in the Supplement. The GHG data can be found at \url{https://doi.org/10.1594/PANGAEA.871273}. The combined ice-sheet, as well as the separate Northern Hemispheric and Greenland ice-sheets are also publicly available on the Research Data Australia repository
at http://handle.unsw.edu.au/1959.4/resource/collection/resdatac_874/1 and doi \textbf{to be added on June 21}. In addition, all the forcing files as well as the paleo-data described in the manuscript are available on the PMIP4 wiki: \url{https://pmip4.lsce.ipsl.fr/doku.php/exp_design:degla_t2}."

---

## Author Response (AR2)

Dear Dr. Hargreaves,

Thank you for your time spent editing our manuscript. Please find answers to your comments in blue below as well as changes that have been made into the manuscript in green.

It is expected for model experiment description papers at GMD to show some evidence that the experiment described works as expected. Usually this involves including some evidence that the experiment, or a very similar experiment, has been performed in a similar complexity model with an indication of the level of model performance. Sometimes an experiment with a simpler model may suffice. Have I missed it - I can't see that any model runs at all been performed for the PDG? If there are actually some results, please cite them, and briefly discuss their performance. If not, perhaps we can consider that last DG model experiment results are a sufficient proof of concept? In either case I would like to see a subsection somewhere that summarises the key differences between the two experiments in terms of forcing and boundary conditions. You already have a paragraph in the Introduction that highlights the observed differences between the two time intervals, but that is not quite the same thing... Also, would it be possible to include some indication of model output in Figure 1 (or elsewhere) for either deglaciation?

No transient simulation of the penultimate deglaciation has been published yet with AOGCMS. This is also why our manuscript is quite timely, as it will allow all transient simulations to follow (or comment on) the protocol detailed here.

The Introduction already includes a lot of information on the experiments that have already been performed for part of the penultimate deglaciation, the LIG or the last deglaciation:

p5, L.21-30, describes snapshot experiments of 130ka and 126ka that have been performed with AOGCMs and models of intermediate complexity. It highlights the limits of these experiments and the need to perform transient simulations.

p5, L. 31-33, describes the transient experiments of the last deglaciation that have been performed with AOGCMs and models of intermediate complexity. These experiments have been quite successful, which is also why a PMIP4 working group on deglaciations has been set-up.

p5, L. 33-35, details the transient simulations that have been performed for the period 135-115 ka with models of intermediate complexity.

To complement this we have now added a sentence within the paragraph p5, L. 33:

"Currently, no 3-dimensional transient simulations were run across the full time interval covered by the penultimate deglaciation (~140-127ka). However, transient simulations covering the period 135 to 115 ka have been performed with a range of models to understand the impact of surface boundary conditions and freshwater fluxes on the LIG (Bakker et al., 2013, Loutre et al., 2014, Goelzer et al., 2016). "

Differences between the penultimate and last deglaciations are highlighted throughout the manuscript, and particularly in the introduction: from p3, L. 24 to p5, L. 14.

To highlight the proof of concept, we have added at the beginning of section 6:

"To this date, no transient simulation covering the period 140 to 127 ka has been performed with a 3-dimensional climate model. However, the climate evolution over the period 135 to 115 ka has been successfully simulated with the Earth system model LOVECLIM (Loutre et al. 2014, Goelzer et al., 2016). Their simulations highlight the potential and feasibility of transient simulations of the PDG. In addition, as a proof of concept, transient experiments of the last deglaciation have been successfully performed with AOGCMs and Earth system models (Liu et al., 2009, Menviel et al., 2011, Roche et al., 2011, Gregoire et al., 2012, He et al., 2013, Otto-Bliesner et al., 2014). As detailed in the previous sections,the proposed transient simulation of the PDG will use boundary forcings consistent with the ones used for the last deglaciation (Ivanovic et al., 2016): i.e. appropriate orbital parameters, greenhouse gases concentration, continental ice-sheets geometry and

meltwater input (Figure 1). To maximise the use of the transient simulations, the transient simulations of the penultimate and last deglaciations, as well as the piControl should be performed with the same version of the climate model."

Since no transient simulation of the PDG has yet been performed, we do not think it is appropriate, nor that it would add value to the manuscript, to show outputs from previous simulations of the last deglaciation. Please note that no such figures were included in the PMIP4 LGM (Kageyama et al., 2017) or last deglaciation (Ivanovic et al., 2017) manuscripts.

Please also update the code and data availability section with the correct unique identifiers for the data sets.
The doi of the dataset has been added to the data availability section.

Apart from these two hopefully small things the manuscript looks ready to be published.
Thank you for considering our manuscript for publication in Geoscientific Model Development.

[revised manuscript text omitted]